# The multiscale brain structural re-organization that occurs from childhood to adolescence correlates with cortical morphology maturation and functional specialization

Yirong He[1], Debin Zeng[2], Qiongling Li[1,3,4], Lei Chu[2], Xiaoxi Dong[1], Xinyuan Liang[1,3,4], Lianglong Sun[1,3,4], Xuhong Liao[5], Tengda Zhao[1,3,4], Xiaodan Chen[1,3,4], Tianyuan Lei[1,3,4], Weiwei Men[6,7], Yanpei Wang[1], Daoyang Wang[1,8], Mingming Hu[1], Zhiying Pan[1], Haibo Zhang[1], Ningyu Liu[1], Shuping Tan[9], Jia-Hong Gao[6,7,10], Shaozheng Qin[1,3,4,11], Sha Tao[1], Qi Dong[1], Yong He[1,3,4,11], Shuyu Li [ID][1]*

1 State Key Laboratory of Cognitive Neuroscience and Learning, Beijing Normal University, Beijing, China, 2 Beijing Advanced Innovation Center for Biomedical Engineering, School of Biological Science & Medical Engineering, Beihang University, Beijing, China, 3 Beijing Key Laboratory of Brain Imaging and Connectomics, Beijing Normal University, Beijing, China, 4 IDG/McGovern Institute for Brain Research, Beijing Normal University, Beijing, China, 5 School of Systems Science, Beijing Normal University, Beijing, China, 6 Center for MRI Research, Academy for Advanced Interdisciplinary Studies, Peking University, Beijing, China, 7 Beijing City Key Laboratory for Medical Physics and Engineering, Institute of Heavy Ion Physics, School of Physics, Peking University, Beijing, China, 8 Zhejiang Philosophy and Social Science Laboratory for Research in Early Development and Childcare, Hangzhou Normal University, Hangzhou, China, 9 Beijing Huilongguan Hospital, Peking University Huilongguan Clinical Medical School, Beijing, China, 10 IDG/McGovern Institute for Brain Research, Peking University, Beijing, China, 11 Chinese Institute for Brain Research, Beijing, China

* shuyuli@bnu.edu.cn

## Abstract

From childhood to adolescence, the structural organization of the human brain undergoes dynamic and regionally heterogeneous changes across multiple scales, from synapses to macroscale white matter pathways. However, during this period, the developmental process of multiscale structural architecture, its association with cortical morphological changes, and its role in the maturation of functional organization remain largely unknown. Here, using two independent multimodal imaging developmental datasets aged 6–14 years, we investigated developmental process of multiscale cortical organization by constructing an in vivo multiscale structural connectome model incorporating white matter tractography, cortico–cortical proximity, and microstructural similarity. By employing the gradient mapping method, the principal gradient derived from the multiscale structural connectome effectively recapitulated the sensory-association axis. Our findings revealed a continuous expansion of the multiscale structural gradient space during development, characterized by enhanced differentiation between primary sensory and higher-order transmodal regions along the principal gradient. This age-related differentiation paralleled regionally heterogeneous changes in cortical morphology. Furthermore, the developmental changes in coupling between multiscale structural and functional connectivity were

**Data availability statement:** The data and source code were available at Zenodo (https://zenodo.org/records/14874537) and GitHub repository (https://github.com/LSYLAB/Multiscale-structural-gradient-differentiation-from-childhood-to-adolescence). Geodesic distance was calculated by the imGeodesics toolbox (https://github.com/mattools/matImage/wiki/imGeodesics). Multiscale structural connectome was generated using the package provided at https://github.com/MICA-MNI/micaopen/tree/master/structural_manifold. Gradient analysis was analyzed by the BrainSpace toolbox (https://github.com/MICA-MNI/brainspace). Partial least square correlation was performed with the myPLS toolbox (https://github.com/danizoeller/myPLS). The AHBA dataset was publicly available at https://human.brain-map.org/static/download and genetic data were preprocessed with abagen toolbox (https://github.com/netneurolab/abagen). The variogram matching approach was used to estimate the spatial correlation significance by generating surrogate maps (https://github.com/murraylab/brainsmash).

**Funding:** The study was supported by the Scientific and Technological Innovation (STI) 2030-Major Projects 2021ZD0200500 (https://en.most.gov.cn/), the National Natural Science Foundation of China (https://www.nsfc.gov.cn/english/site_1/index.html, 32271146 to SL, 82021004 to YH, 82202245 to QL, 31521063 to QD, ST, YW, DW, MH, and ZP), the Startup Funds for Top-notch Talents at Beijing Normal University to SL, and Beijing Municipal Science and Technology Commission (https://kw.beijing.gov.cn/, Z15110000391512 to ST, YW, DW, MH, and ZP). The funders had no role in study design, data collection and analysis, decision to publish, or preparation of the manuscript.

**Competing interests:** The authors have declared that no competing interests exist.

**Abbreviations:** AHBA, Allen Human Brain Atlas; ANT, Attention Network Test; BIC, Bayesian information criterion; CBD, Children School Functions and Brain Development Project in China (Beijing Cohort); DAN, dorsal attention network; DMN, default mode network; EC, executive control; FC, functional connectivity; FDR, false discovery rate; FPN, frontoparietal network; GAMM, generalized additive mixed model; GC, Gaussian curvature; GD, geodesic distance; GO, gene ontology; HCP, Human Connectome Project; LC, latent component; MC, mean curvature; MPC, microstructural profile covariance; PaC, participation

correlated with functional specialization refinement, as evidenced by changes in the participation coefficient. Notably, the differentiation of the principal multiscale structural gradient was associated with improved cognitive abilities, such as enhanced working memory and attention performance, and potentially underpinned by synaptic and hormone-related biological processes. These findings advance our understanding of the intricate maturation process of brain structural organization and its implications for cognitive performance.

## Introduction

The human brain is a complex network that follows coordinated structural organizational principles at multiple spatial scales [1]. From microscale neuron-to-neuron interactions to macroscale anatomical pathways connecting different brain regions, the anatomical connections encompass a range of scales [2–4]. Multiscale structural organization emerges from complex biological mechanisms and serves as a foundational framework to support various brain functions [1, 2]. Reconstructing the human brain structural connectome across multiple scales has implications for comprehending the principles of human brain organization and the foundation of cognitive function.

To comprehensively characterize neural organizations across multiple scales, an in vivo structural wiring model integrating complementary neuroimaging features based on multimodal magnetic resonance imaging (MRI) has recently been proposed [5]. These features include macroscale structural characteristics, encompassing diffusion MRI tractography, cortical geodesic distance (GD), and microscale structural features called microstructural profile covariance (MPC) [5]. Diffusion MRI (dMRI) tractography, while prevalent for inferring deeper white matter tracts, exhibits limitations in detecting connections within gray matter and superficial white matter fibers, and long-distance projections are systematically under-recovered [6, 7]. The GD captures spatial proximity of the cortex which may estimate wiring cost, enabling estimation of short-range connections in gray matter [8] and superficial white matter. Moreover, microstructural similarity is linked to connection probability in non-human animals, and the cortico-cortical "structural model" posits a close association between connectivity likelihood and similarity of microstructure across cortical regions [9, 10]. The MPC assesses this similarity between cortical regions by systematically comparing the myelin-sensitive neuroimaging profiles in cortical columns [11]. By incorporating GD and microstructural similarity as additional structural connectivity features, this multiscale structural model compensates for the limitations of relying solely on dMRI tractography [5]. By employing the gradient mapping technique, previous studies revealed the existence of the principal organizational axis derived from the multiscale structural connectome in healthy adults [5] and individuals aged 14–25 years [12]. Remarkably, this principal organizational axis spatially aligns with the principal axis of large-scale cortical organization known as the "sensorimotor-association (S-A) cortical axis" [13, 14]. This axis represents continuous transitions of cortical properties across the cortical mantle from primary to association regions, capturing a hierarchical organization that manifests in anatomy [15], function [14], and evolution [16].

Childhood and adolescence (6–14 years of age) represent a critical period of rapid and continuous brain development marked by the restructuring of neural circuits influenced by puberty hormones. This restructuring leads to permanent brain structural reorganization and significant gains in cognitive and emotional functions, with a cognitive transition from concrete to abstract and logical thinking [17–19]. Concurrently, the functional organization of the brain undergoes significant reconfigurations, with the principal axis shifting from a

coefficient;PCA, principal component analysis; PLSC, partial least square correlation; PLSR, partial least squares regression; S-A, sensorimotor-association; SA, Surface area; SN, somatomotor network; TS, tract strength; VAN, ventral attention network; VN, visual network;WM, working memory.

visual-sensorimotor gradient to a pattern gradient delineated by the S-A axis [20, 21]. This period is also characterized by dynamic and regionally heterogeneous changes in brain structural features across multiple scales. For example, there are pronounced changes at the microscale level, including the growth of intracortical myelination and synaptic pruning [22, 23]. Moreover, the maturation of white matter leads to a substantial reorganization of large-scale brain structural networks [24, 25]. Consequently, delineating the development of multiscale structural organization during this period can yield structural insights into the significant functional reorganization and cognitive development.

From childhood to adolescence, cortical morphology undergoes remarkable refinements, including cortical surface area expansion and cortical thinning [26–28]. Previous studies associated cortical morphology with multiscale structural connectivity, revealing that regions with similar morphological features were more likely to exhibit axonal connectivity and to share comparable cytoarchitecture [29, 30]. In addition, biological processes potentially linked to the refinement of multiscale structural wiring architecture, such as microscale myelin proliferation into the periphery of the cortical neuropil, dynamic synapse reorganization, macroscale white matter fiber development, and axonal mechanical tension, are hypothesized to contribute to the maturation of cortical morphology [31–36]. Thus, the potential association between the development of multiscale structural gradients and regionally heterogeneous maturation of cortical morphology warrants further exploration. Furthermore, although dynamic functional interactions between brain regions are constrained by invariant multiscale structural wiring, divergence between structural and functional networks may support flexible and diverse cognitive functions [1,37]. Corresponding to the development of structural brain networks, large-scale functional networks exhibit a shift toward a more segregated network topology, facilitating flexible and specialized brain functions [38–41]. Therefore, it is worthwhile to investigate how structural constraints contribute to the maturation of functional organization and cognitive development. In addition, accumulating evidence indicates that genetic factors closely regulate the development of brain structure across regions [42]. Axon guidance, which is closely linked to the formation of neural circuits during neural development, is associated with structural wiring [43–45]. Therefore, investigating associated gene expression can reveal the underlying biological mechanisms driving multiscale structural development processes.

In this study, we utilized multi-modal neuroimaging data including dMRI, T1-weighted (T1w) MRI, T2-weighted (T2w) MRI, and resting-state functional MRI (rs-fMRI) from independent longitudinal dataset (437 scans, 276 subjects, discovery dataset) and cross-sectional dataset (290 subjects, replication dataset) of children aged 6–14 years. Using the gradient mapping algorithm and generalized additive mixed models (GAMMs), we characterized the developmental patterns of multiscale structural gradients during childhood and adolescence. Furthermore, we explored the associations of these gradients with the refinement of cortical morphology. We also examined the associations between multiscale structure–function coupling and the maturation of cortical organization. Moreover, we investigated the underlying genetic basis and examined the relationships between multiscale structural gradients and individual cognition.

## Results

### Age-related changes in multiscale structural gradient during development revealed the gradual maturation of the S-A axis

The discovery dataset was from the Children School Functions and Brain Development Project in China (Beijing Cohort) (CBD). For the discovery dataset, the multiscale structural gradients were computed for each scan utilizing a complementary model that integrated 3

cortical structural connectivity features (GD, MPC, and dMRI tractography) on a Schaefer 1000 parcellation [46]. Using the diffusion map embedding algorithm, we derived a set of components ranked by the variance they accounted for (Fig 1A, middle panel). Our first and third gradients aligned with patterns observed in previous studies of individuals aged 14–25 years and adults [5,12], with the principal gradient differing between the primary regions (somatomotor network [SN] and visual network [VN]) (positive values) and transmodal regions (default mode network [DMN]) (negative values), reflecting the hierarchical organization of the cortex. The third gradient demarcated the anterior and posterior cortex. While

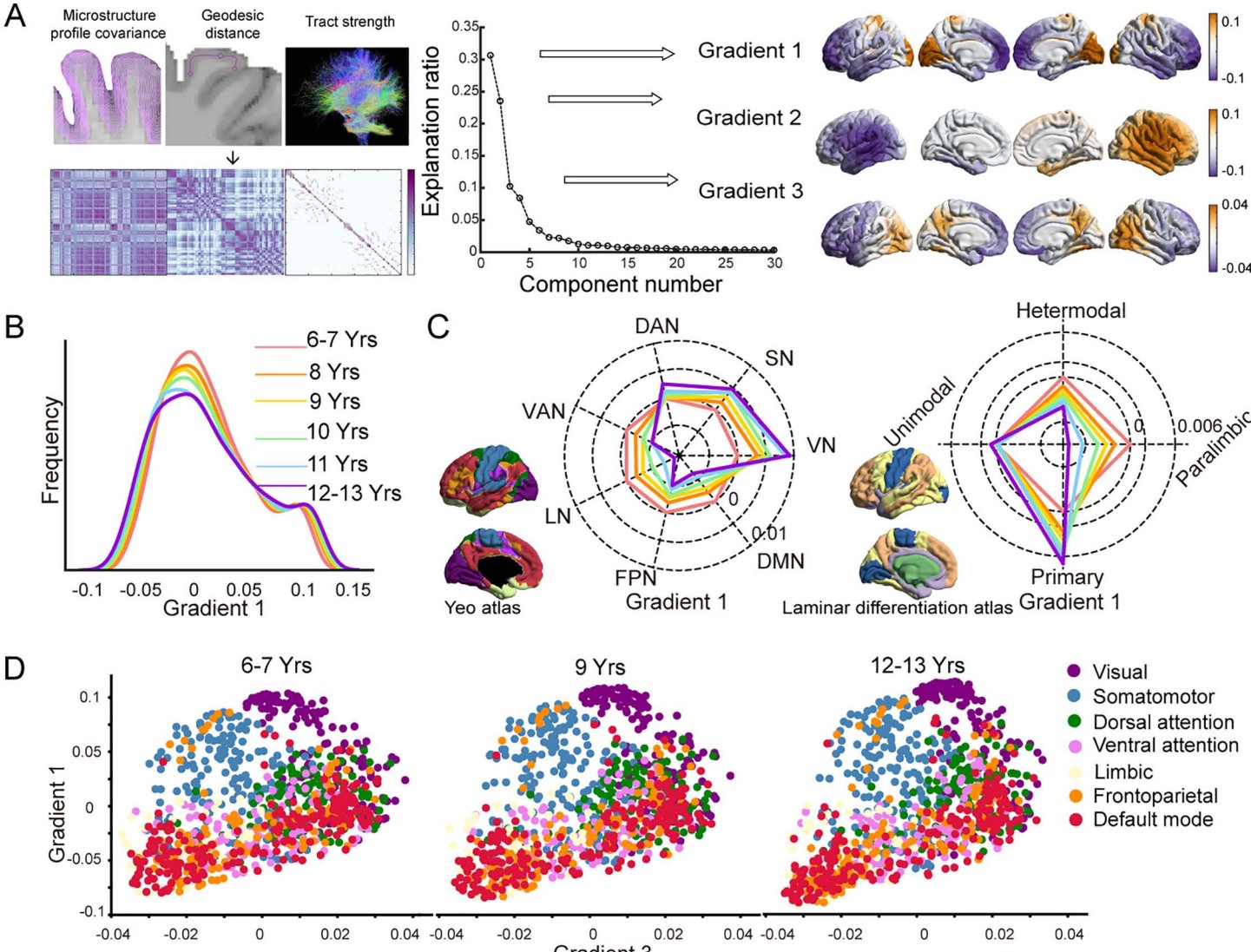

**Fig 1. Multiscale structural gradients during childhood and adolescence.** (A) The matrices containing the structural features of geodesic distance, microstructural profile covariance, and diffusion MRI tractography were concatenated and transformed into an affinity matrix, followed by the diffusion map embedding algorithm. The first three gradients capture the largest proportion of the variance. The group-averaged gradients were projected onto the cortical surface and visually represented (right). (B) The global density map of the principal gradient for six age-specific groups. (C) Radar plot of the principal gradient for comparison between the 6–7-year-old group and other age-specific groups based on Yeo functional networks (left) [47] and laminar differentiation parcellation (right) [48]. (D) The first and third structural gradients mapped into a 2D gradient space for the 6- to 7-, 9-, and 12- to 13-year-old groups. The data underlying this figure can be found at https://zenodo.org/records/14874537.

the second gradient separated left and right hemisphere (Fig 1A, right panel). We focused on the first and third gradients, as they collectively accounted for a substantial proportion (approximately 42%) of the variance in cortical connectivity and specific biological information they compassed. To demonstrate the overall pattern of age-related changes in gradients, we computed group-averaged gradients for six age groups (6–7, 8, 9, 10, 11, and 12–13 years) and compared their global distributions. The group-averaged gradient maps for each group were shown in S2 Fig. Our results demonstrated that from the 6–7 age group to the 12–13 age group, the range of the principal gradient expands bilaterally (Fig 1B). Subsequently, we summarized the principal gradient at the network level according to intrinsic functional communities [47] and the atlas of laminar differentiation [48], as illustrated in Fig 1C. The network analysis also demonstrated an age-related expansion of the principal gradient, manifested by increased gradient scores in regions with high gradient scores (SN, VN) and the dorsal attention network (DAN), while scores decreased in regions with low gradient scores (ventral attention network [VAN], limbic network, frontoparietal network [FPN], and DMN). These findings were corroborated by results from the laminar differentiation atlas (Fig 1C, right panel). The results of the third gradient were shown in S3A, S3B, and S3C Fig. We next constructed a 2-dimensional gradient space using the principal and third gradient to qualitatively assess global distribution patterns in the 6–7-year-old, 9-year-old, and 12–13-year-old groups, as depicted in Fig 1D. The gradient space demonstrated an expansion trend throughout development (the developmental process of the gradient space across different ages was depicted in S1 Movie). Similar observations were also documented in the Schaefer 400 atlas (S4 Fig).

To evaluate the reproducibility of our findings, we incorporated an independent dataset ("replication dataset") comprising cross-sectional multi-modal MRI data from 290 typically developing participants (6–14 years old) from the Lifespan Human Connectome Project in Development (HCP-D) [49]. The results of the replication dataset were consistent with those of the discovery dataset (S5A, S5B, S5C, and S5F Fig and S2 Movie).

To quantify the effect of age on multiscale structural gradients during development, GAMMs were constructed, with age as a smooth term, subject ID as a random effect, and sex and mean framewise displacement (mFD) as linear covariates. We first computed several global measures to describe the overall characteristics of the first and third gradients, including the explanation ratio, range, and standard deviation. The explanation ratio quantified the percentage of connectivity variance accounted for by that gradient, and a higher value signified a more prominent role in the organization of the structural connectome. The range indicated differentiation between the regions at the gradient extremes, and the standard deviation measured the heterogeneity of the gradient scores across regions. We assessed the magnitude of age effects using the variance explained by age ($\Delta$Adj $R^2$) and determined the direction of the age effect by the sign of the age coefficient from the equivalent linear model. We observed age-related increases in the principal gradient (age-effect $p < 0.001$) and decreases in the third gradient (age-effect $p < 0.001$) for all three global measures (Fig 2A). Additionally, dispersion was calculated by summing the Euclidean distances between each point and the centroid within the 2D space formed by the first and third gradients for each individual, providing a quantification of the overall dissimilarity within the gradient space. The gradient dispersion exhibited an increasing pattern during development ($\Delta$Adj $R^2 = -0.17$, $p = 1.48 \times 10^{-19}$) (Fig 2B). These findings indicated a shift toward a more distributed structural network topology during development, with the principal gradient increasingly differentiating between primary and transmodal regions. This finding was consistent with the increasing dominance of the principal gradient. In contrast, the third gradient suggested a progressive weakening of the anterior-posterior pattern.

To examine the statistical age effect across the whole brain, we also leveraged the GAMMs at the node level. We characterized developmental trajectories of the principal gradient in each brain region using age-smooth functions generated by each regional GAMM. Our analysis revealed a continuum of developmental trajectories that varied based on each region's position on the principal gradient rank (Fig 2C, upper panel). Lower-ranking regions (purple) demonstrated a linear decrease in gradient scores, while higher-ranking regions (orange) showed an increase in gradient scores throughout development. The lower panel of Fig 2C summarized the age-effect on principal gradient across the brain regions on the surface, highlighting increases in the SN and VN which corresponding to the positive extremum, while regions linked to the negative extremum, such as the temporal, medial, and lateral prefrontal lobes, exhibited a pattern of decline ($p < 0.05$, Bonferroni corrected). These results indicated a strengthened differentiation of the principal gradient, which corresponded to the S-A axis. To further examine whether the S-A pattern of the principal gradient strengthened during development, we used the primary-to-transmodal functional gradient derived from the group-averaged functional connectivity (FC) matrix as the S-A axis (Fig 2D). We computed

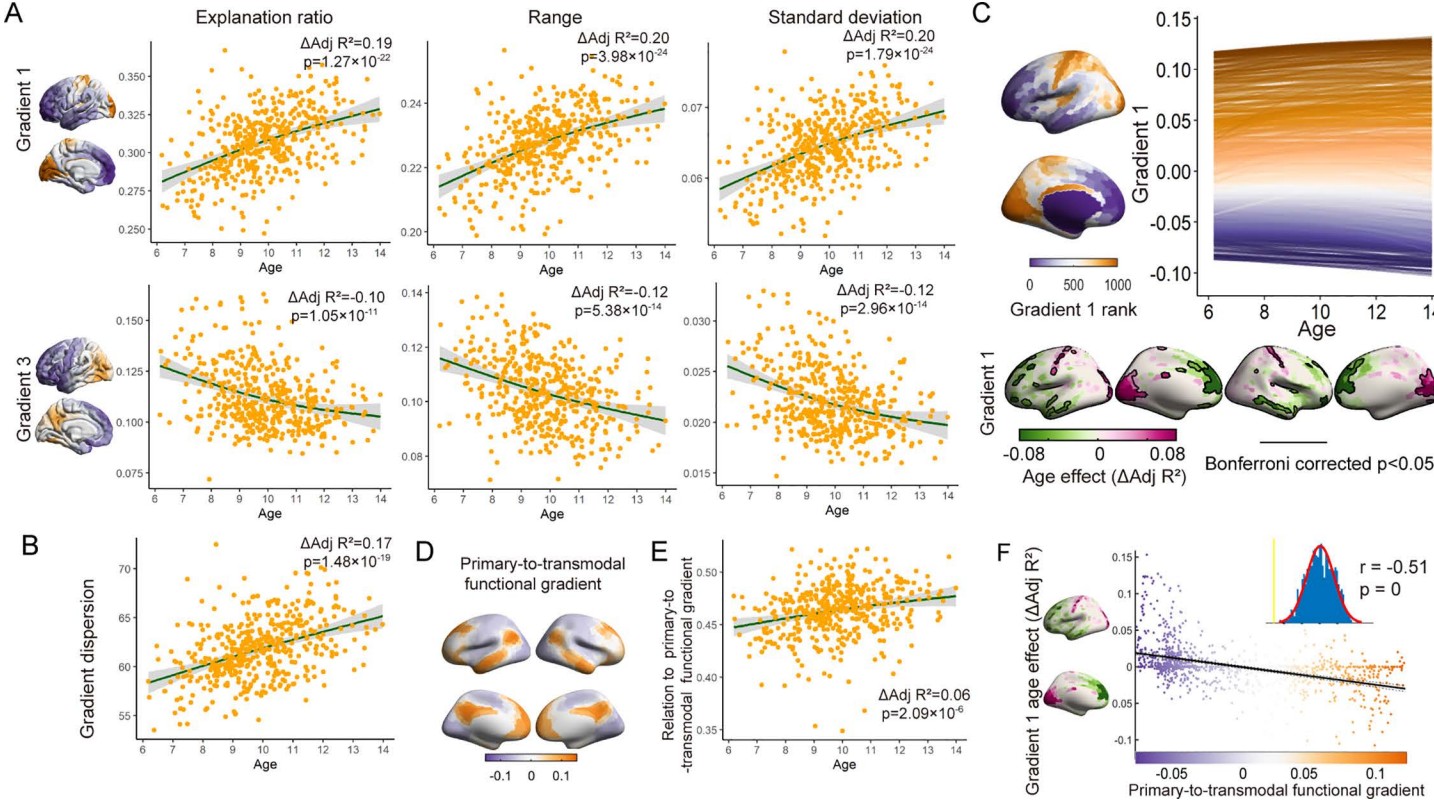

**Fig 2. Age-related changes in gradients at both the global level and the region level. (A)** Global measures of the first and third gradients changed across age groups, including the explanation ratio (left), range (middle), and standard deviation (right) (age-effect $p < 0.01$). **(B)** Age-related changes in gradient dispersion computed from the first and third gradients (age-effect $p < 0.01$). **(C)** Upper panel: The multiscale structural principal gradient developmental trajectories exhibited continuous variation along the gradient rank. Each line illustrated the developmental trajectory of gradient scores for an individual region, modeled using GAMMs. Colors denote the rank of a given region along the gradient. Lower panel: The spatial pattern of age-effect on the principal gradient, with the age effect quantified using $\Delta$Adj $R^2$. The results that survived Bonferroni correction were circled by black lines (Bonferroni corrected $p < 0.05$). **(D)** The primary-to-transmodal functional gradient derived from the group-averaged functional connectivity matrix. **(E)** Age-related changes in the correlation coefficient between the multiscale structural principal gradient and the primary-to-transmodal functional gradient (age effect $p < 0.01$). **(F)** Spatial correlation between the structural principal gradient age-related $\Delta$Adj $R^2$ map and the primary-to-transmodal functional gradient. Each dot represented a brain region. The significance level was corrected for spatial autocorrelation ($p_{surrogate} < 0.01$). The data underlying this figure can be found at https://zenodo.org/records/14874537.

the correlation coefficient between the principal structural gradient and functional gradient for each scan. The GAMMs revealed a significant increase in the correlation coefficient during development, which indicated a strengthened S-A pattern in multiscale structural organization ($\Delta$Adj $R^2 = 0.06$, $p = 2.09 \times 10^{-6}$) (Fig 2E). In addition, the age-related $\Delta$Adj $R^2$-map of the multiscale structural principal gradient demonstrated a significant correlation with the primary-to-transmodal gradient, indicating temporal changes following the S-A organization pattern ($r = -0.51$, $p_{\text{surrogate}} = 0.00$, corrected for spatial autocorrelation) (Fig 2F). Therefore, this analysis demonstrated that multiscale structural wiring architecture shifted toward a more distributed hierarchical organization during childhood and adolescence. These results were also validated in the replication dataset (S5D, S5E, and S5G Fig).

## Multiscale structural principal gradient and its maturation are associated with the development of cortical morphology

We hypothesized that the refinement of the multiscale structural principal gradient may align with the heterogeneous maturation of cortical morphology, as cortical regions with similar morphological features are more likely to have structural connections. Subsequently, we employed five cortical morphometric measures, including cortical thickness, gray matter volume, surface area, mean curvature (MC), and Gaussian curvature (GC), to create a comprehensive cortical morphological profile. We investigated the associations between the multiscale structural principal gradient and morphometric features (Fig 3A). Given the similarities in the spatial patterns of these metrics, we performed principal component analysis (PCA) to project the five features onto a set of principal axes that effectively captured the spatial variation in the cortical morphological profile. The first component (PC1) explained nearly 85% of the variance, and we incorporated PC1 into subsequent analyses. As shown in Fig 3B, the group-averaged PC1 exhibited differentiation between primary regions (i.e., the SN and VN) and transmodal regions (i.e., the FPN and DMN), indicating that distinct morphometric attributes distinguish these two types of brain regions. Then, as depicted in Fig 3C, we explored the relationship between PC1 and multiscale structural gradient 1 and identified a strong correlation ($r = -0.78$, $p_{\text{surrogate}} = 0.00$, corrected for spatial autocorrelation). These findings suggested a potential association between cortical morphology and cortical wiring architecture across the cortical mantle, as regions exhibiting similar morphological features also display comparable multiscale structural connectivity profiles.

To validate the presence of a developmental association between cortical wiring and cortical morphology, we investigated the spatial correlation of mature patterns between them. Specifically, we employed the GAMMs on PC1 to characterize the effect of age on cortical morphology. As illustrated in the right panel of Fig 3D, we observed an increase in the prefrontal lobe, which occupies the positive end of PC1. This observation suggested distinct maturation processes between the prefrontal lobe and other brain regions. Moreover, as shown in the left panel of Fig 3D, the correlation analysis between the $\Delta$Adj $R^2$-maps of multiscale structural gradient 1 and morphometric PC1 revealed a congruent developmental pattern with a correlation coefficient of $r = -0.40$ ($p_{\text{surrogate}} = 0.003$, corrected for spatial autocorrelation). The increase in the multiscale structural principal gradient in the SN was accompanied by a decrease in PC1, while the decrease in the principal gradient in the prefrontal and temporal lobes was accompanied by an increase in PC1. The obtained results validated our hypothesis that there are synchronized maturation patterns between cortical wiring and cortical morphology. As shown in Fig 3E, to investigate the extent to which individual morphological features co-evolve with the multiscale structural gradient, we also conducted a correlation analysis between the $\Delta$Adj $R^2$-map of multiscale structural gradient 1 and the $\Delta$Adj $R^2$-map of each morphometric feature. Notably, a significant association was observed between $\Delta$Adj

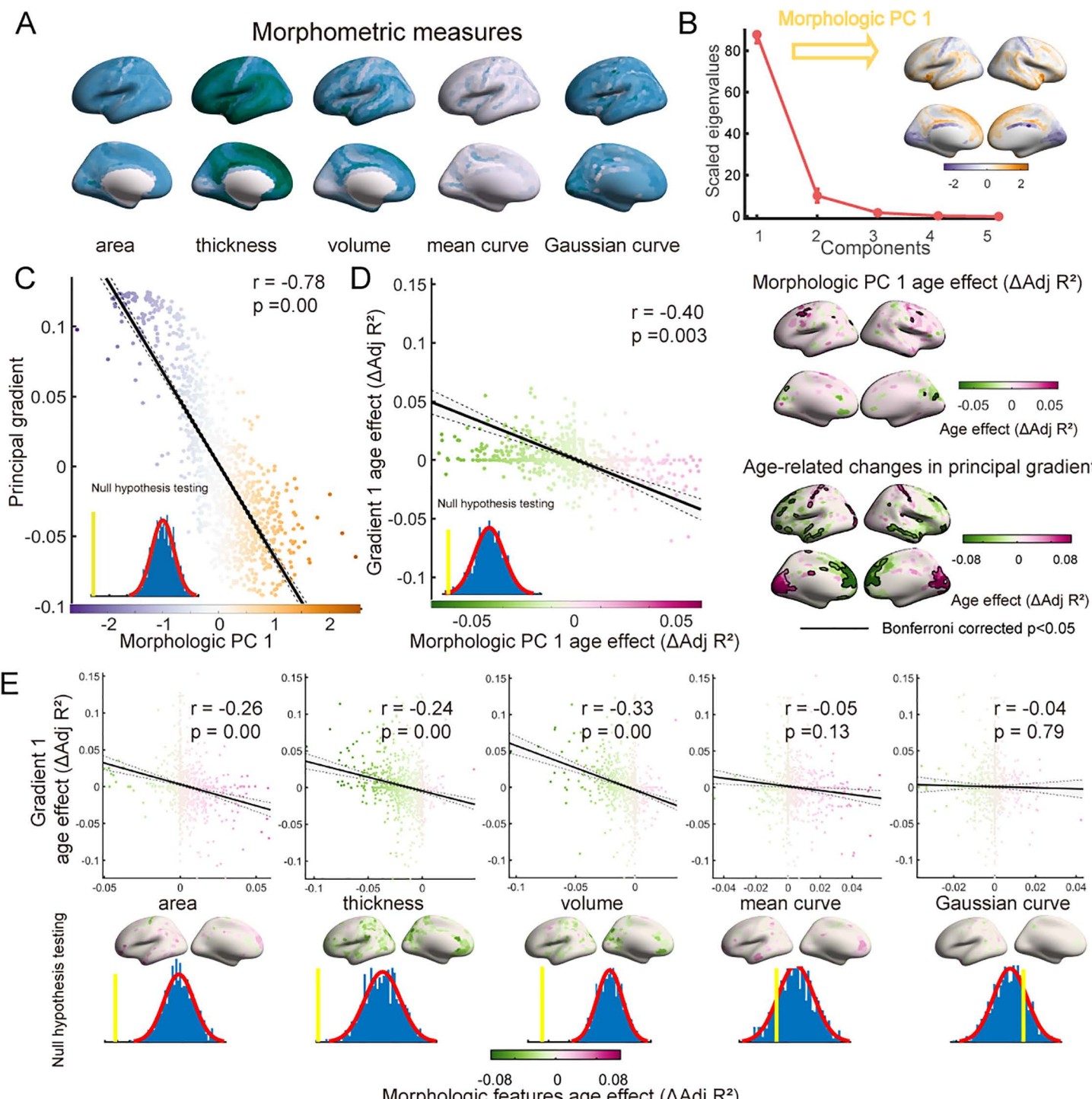

**Fig 3. Association between the multiscale structural principal gradient and morphometric features. (A)** Group-averaged morphometric features, including cortical thickness, gray matter volume, surface area, mean curvature, and Gaussian curvature. **(B)** The five morphometric features were input into the PCA algorithm, and components were ordered according to the proportion of variance they accounted for. The principal component (PC1) was mapped on the surface (right). **(C)** Spatial correlation between the multiscale structural principal gradient and morphometric PC1. Each dot represents a brain region. The significance level was corrected for spatial autocorrelation ($p_{\text{surrogate}} = 0.00$). **(D)** Spatial correlation between age-related $\Delta$Adj $R^2$-maps of the multiscale structural principal gradient and morphometric PC1 ($p_{\text{surrogate}} = 0.003$). **(E)** Spatial correlation between age-related $\Delta$Adj $R^2$-maps of the multiscale structural principal gradient and morphometric features, including surface area, cortical thickness, gray matter volume, mean curvature, and Gaussian curvature. The data underlying this figure can be found at https://zenodo.org/records/14874537.

$R^2$-maps of the principal gradient and surface area ($r = -0.26$, $p_{surrogate} = 0.00$), cortical thickness ($r = -0.24$, $p_{surrogate} = 0.00$), and gray matter volume ($r = -0.33$, $p_{surrogate} = 0.00$). The results in the replication dataset were largely consistent with these findings (S8 Fig).

Based on the five morphometric features, we constructed morphometric similarity networks (MSN) for each subject (S9 Fig). There is a significant correlation between the edges of MSN and the edges of multiscale structural connectivity (S9B Fig). Furthermore, MSN principal gradient was also significantly correlated with multiscale structural gradient 1 ($r = -0.65$, $p_{surrogate} = 0.00$) (S9C Fig). The ΔAdj $R^2$-maps of multiscale structural gradient 1 and MSN gradient 1 also exhibited a significant spatial correlation ($r = -0.16$, $p_{surrogate} = 0.00$) (S9D Fig).

Altogether, these findings provided evidence of interconnected spatial patterns and developmental influences between the multiscale structural connectome and cortical morphology.

## Development of multiscale structure–function coupling associated with the refinement of cortical functional specialization

The coupling between structure and function indicates that structure serves as a fundamental framework that for synchronized fluctuations in functional activities underlying cognition [50]. To further investigate the role of the multiscale structural connectome in shaping the development of functional architecture, we assessed the coupling between structure and function for each region using Spearman rank correlation (Fig 4A). As shown in Fig 4B, the group-averaged coupling map revealed distinct patterns across the cortex, reflecting the alignment of functional and multiscale structural connectivity profiles of the given region. The significant spatial correlation coefficient between coupling and multiscale structural principal gradient further revealed a hierarchical pattern across the cortical mantle characterized by greater levels of coupling in primary regions and lower levels in transmodal regions ($r = 0.56$, $p_{surrogate} = 0.00$, corrected for spatial autocorrelation) (Fig 4C). A previous study linked variability in structure–function coupling to functional specialization [50]. In the context of multiscale structural connectivity, we calculated the participation coefficient (PaC) for each node based on both multiscale structural and functional networks to evaluate inter-module connectivity and functional specialization, with lower PaC values indicating lower degree of involving in other modules. The correlation between multiscale structure–function coupling and group-averaged PaC maps was illustrated in Fig 4D, revealing a significant relationship (correlation with structural PaC: $r = -0.66$, $p_{surrogate} = 0.00$; functional PaC: $r = -0.67$, $p_{surrogate} = 0.00$, corrected for spatial autocorrelation). These findings indicated that greater multiscale structure–function coupling was associated with greater functional specialization, while lower coupling corresponded to greater functional integration. These findings aligned closely with the previous study, suggesting that the multiscale structure–function coupling reflected functional specialization and hierarchy [50].

We employed the GAMMs to characterize age-related changes in regional multiscale structure–function coupling. As depicted in Fig 4E, the prefrontal cortex and visual cortex exhibited enhanced coupling during development (Bonferroni corrected $p < 0.05$). Considering the close interplay between multiscale structure–function coupling and segregation, we further hypothesized that age-related changes in coupling are accompanied by alterations in the PaC. As depicted in Fig 4F, the correlation analysis between ΔAdj $R^2$-maps of coupling and structural as well as functional PaCs revealed a congruent developmental pattern (correlation with structural PaC: $r = -0.17$, $p_{surrogate} = 0.01$; functional PaC: $r = -0.17$, $p_{surrogate} = 0.00$, corrected for spatial autocorrelation). This finding suggested that brain regions exhibiting increases in multiscale structure–function coupling were more likely to be accompanied by an increased degree of functional specialization. Taken together, these findings demonstrated that the maturation of multiscale structure–function coupling was related to the refinement of

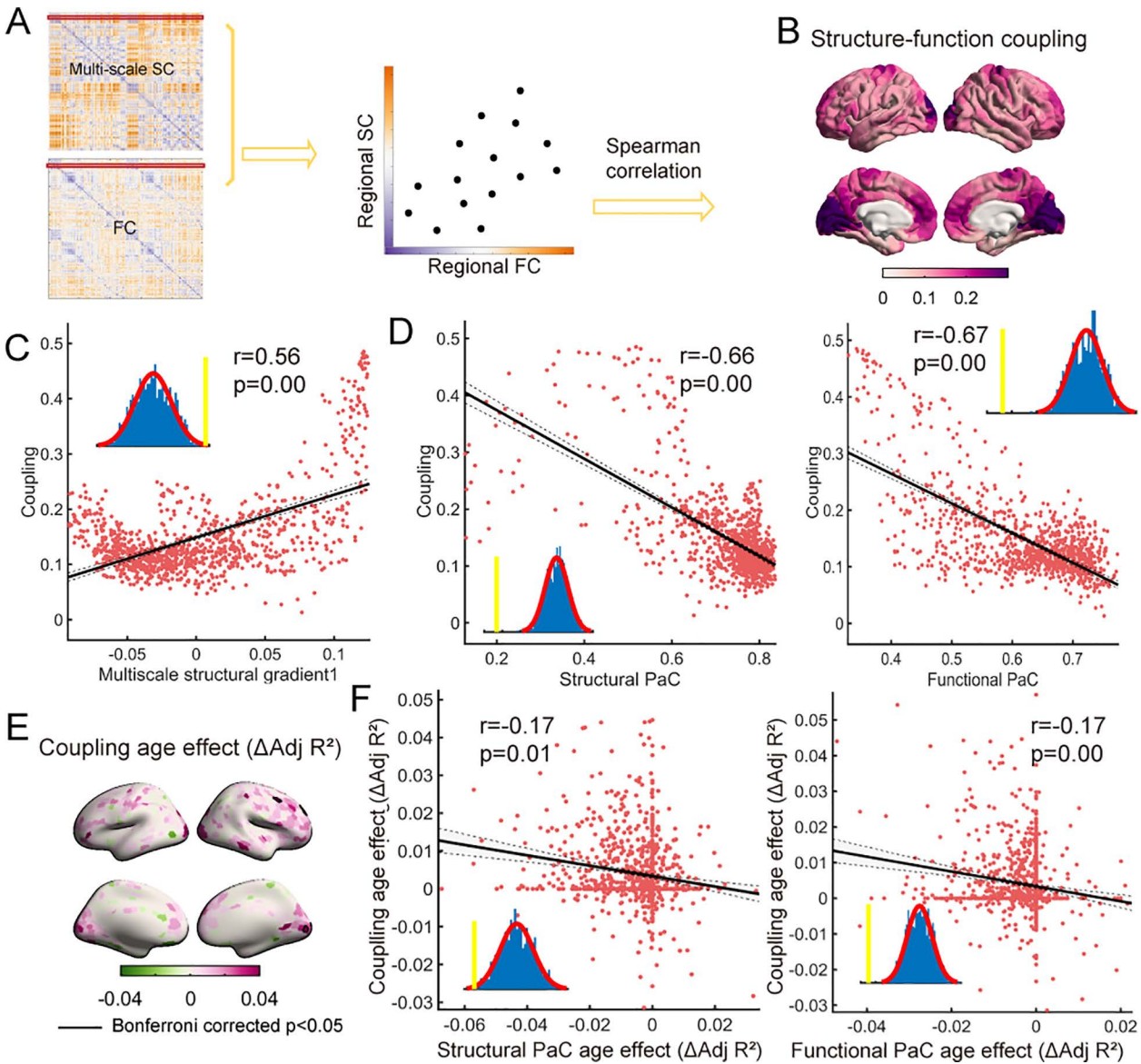

**Fig 4. Multiscale structure–function coupling during development.** (**A**) For each region, multiscale structure–function coupling was calculated as the Spearman correlation coefficient between the multiscale SC and FC profiles of that region. (**B**) A group-averaged multiscale structure–function coupling map of the cortical surface was depicted. (**C**) Spatial correlation between multiscale structure-function coupling map and multiscale structural principal gradient. (**D**) Spatial correlation between the multiscale structure–function coupling map and the structural/functional participation coefficient map. (**E**) Age-related changes in multiscale structure–function coupling, with the age effect quantified using ΔAdj $R^2$. The results that survived Bonferroni correction were circled by black lines (Bonferroni corrected $p < 0.05$). (**F**) Spatial correlation between ΔAdj $R^2$-maps of multiscale structure–function coupling and the structural/functional participation coefficients. The data underlying this figure can be found at https://zenodo.org/records/14874537.

functional specialization from childhood to adolescence. The results in the replication dataset were largely consistent with these findings (S10 Fig).

## Differentiation of the principal multiscale structural gradient was related to better cognitive performance

Structural connectivity forms the fundamental basis for neuronal interactions that underlie the emergence of cognition and behavior [37]. Throughout childhood and adolescence,

attention and executive function undergo continuous enhancement [51]. Subsequently, we sought to explore the implications of cortical wiring for individual cognition by investigating two cognitive dimensions: working memory (WM) and attentional ability. Here, WM was measured by a typical numerical $n$-back task, while attention performance was measured by response time for alerting, orienting, and executive control (EC) tasks (see Methods for further details). We next assessed the associations between the gradient data and cognition data across individuals via partial least square correlation (PLSC) analysis, which captures complex relationships within multidimensional data. Considering the distinct cognitive aspects assessed by the two tests, separate PLSC analyses were performed for each cognitive domain. Through PLSC, we generated latent components (LCs) that captured the optimal associations between the principal gradient and cognitive scores.

For WM, the first LC (LC1) exhibited significance in the permutation test ($p < 0.01$). For LC1, the composite scores were computed by projecting the original data onto their corresponding weights. The correlation between the WM composite scores and the gradient 1 composite scores was significant, indicating a strong positive relationship between the cognitive and gradient data ($r = 0.46$, $p = 0.001$) (Fig 5A). Additionally, we calculated the loadings of gradient 1 and WM by computing the Pearson correlation between the original data and the composite scores, thereby quantifying the contribution of the given brain (cognitive) measure for the LC. As shown in Fig 5B and 5C, higher WM composite scores were associated with worse WM performance, while greater gradient composite scores were linked to higher values of gradient 1 in transmodal regions and lower values in primary regions. These significant loadings, tested by bootstrap resampling ($n = 1,000$), were depicted with shadows in WM and black lines in gradient 1. Better WM performance was associated with higher gradient 1 values in primary regions and lower values in transmodal regions.

Similar to the WM results, LC1 derived from the attention-related PLSC analysis showed a significant association between attention and gradient 1 composite scores ($r = 0.37$, $p = 0.001$) (Fig 5D). As shown in Fig 5E and 5F, better attention scores were associated with higher gradient 1 values in transmodal regions and lower values in primary regions. Given that attention performance was measured through response time, larger attention scores indicated poorer attention performance. Therefore, these findings were consistent with the results obtained from the WM analysis, suggesting a significant association between improved cognitive performance and decreased negative value as well as increased positive value of the principal gradient (strengthened S-A pattern in multiscale structural organization).

The findings from the replication dataset were largely consistent with the original results, encompassing various cognitive measures such as fluid intelligence, episodic memory, executive function/cognitive flexibility, inhibition, language/reading decoding, vocabulary comprehension, processing speed, and WM (S11 Fig). Furthermore, we conducted 10 iterations of 5-fold cross-validation of the PLSC analysis (see Methods for further details). The mean correlation between cognitive and brain scores of test set was (WM: $r = 0.22$, $p = 0.04$; ATT: $r = 0.16$, $p = 0.07$ for discovery dataset; multiple cognitive measures: $r = 0.45$, $p = 0.02$ for replication dataset) (See S1 Table for more details). Collectively, these outcomes provide compelling evidence that the enhancement of the S-A axis pattern along multiscale structural gradient 1 is associated with improved cognitive performance.

## Maturation of the principal multiscale structural gradient was associated with gene expression profiles

To explore the underlying biological mechanisms of the maturation of multiscale structural gradients, we applied genome expression data from the Allen Human Brain Atlas (AHBA)

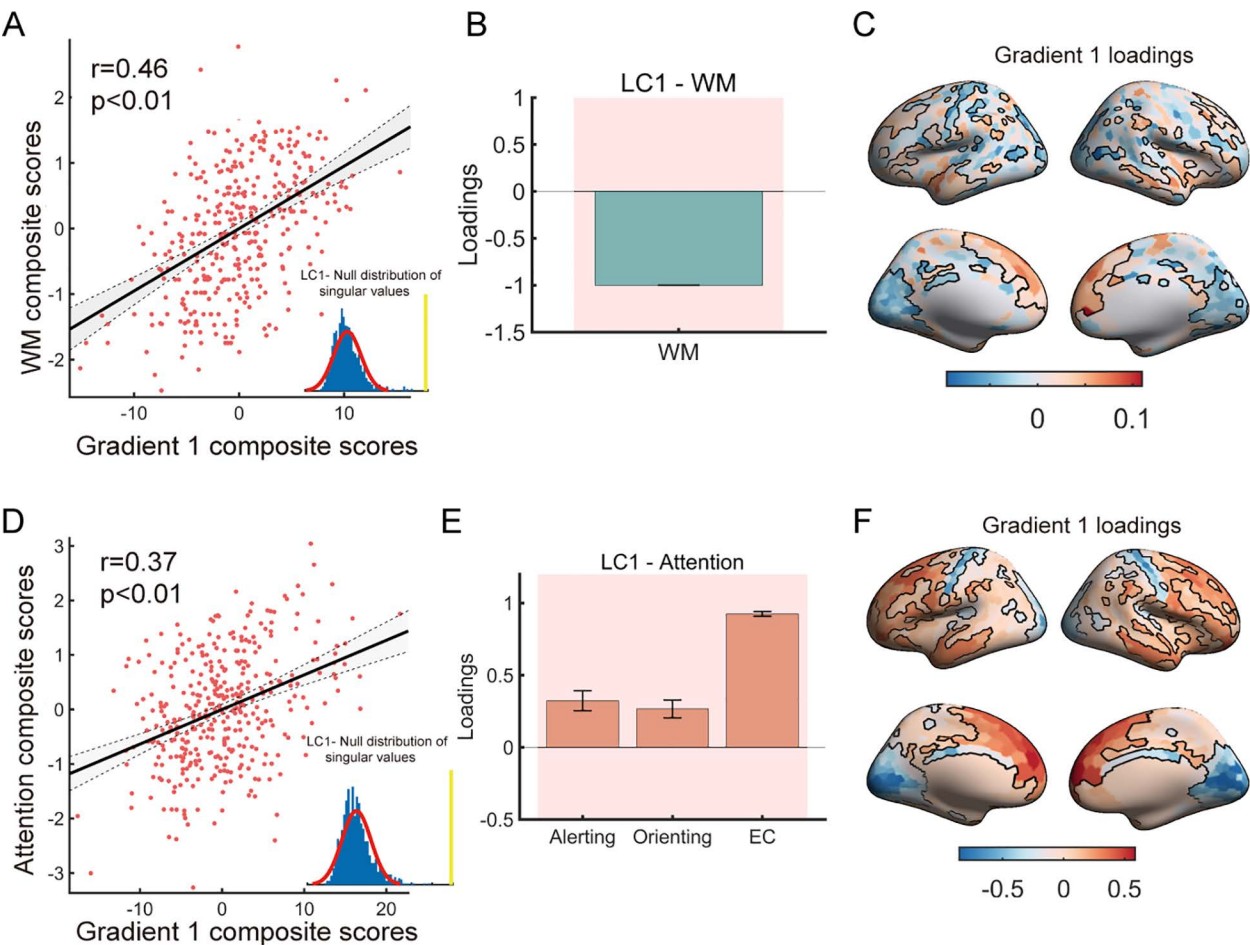

**Fig 5. Partial least square correlation (PLSC) analysis revealed an association between the principal gradient and cognitive scores. (A, D)** Pearson correlations between the principal gradient composite scores and working memory/attention composite scores. The inset figure shows the null distribution of singular values estimated by the permutation test ($n$ = 1,000). **(B, E)** Loadings of WM/attention were calculated by Pearson correlation between the cognitive measurements and their composite scores. The shadows represent significant loadings tested by bootstrap resampling ($n$ = 1,000). **(C, F)** Gradient loadings were calculated by Pearson correlation between gradient 1 and their composite scores. The loadings of regions with black lines were subjected to a significance test by bootstrap resampling ($n$ = 1,000). The data underlying this figure can be found at https://zenodo.org/records/14874537.

(https://human.brain-map.org [52]). The microarray data were preprocessed using the abagen toolbox (version 0.1.3; https://github.com/rmarkello/abagen). Given that data from the right hemisphere were incomplete, we only used the data from the left hemisphere. By mapping the microarray data to the Schaefer 1000 atlas, we obtained a $416 \times 15,631$ (region × gene) matrix (Fig 6A).

Subsequently, we employed a partial least squares regression (PLSR) algorithm to investigate the relationships between the age-related gradient 1 $\Delta$Adj $R^2$-map and the gene expression matrix. The first component (PLS1) accounted for the largest proportion of the variance (49%) and represented the optimally weighted linear combinations of gene expression patterns (Fig 6B). The spatial pattern of PLS1 was spatially correlated with the multiscale structural gradient 1 $\Delta$Adj $R^2$-map ($r = 0.70$, $p_{surrogate} = 0.01$, corrected for spatial autocorrelation) (Fig 6C).

To further investigate the biological implications, the genes were ranked based on the weights from PLS1, and the top 10% of genes from both the positive (PLS1 + ) and negative

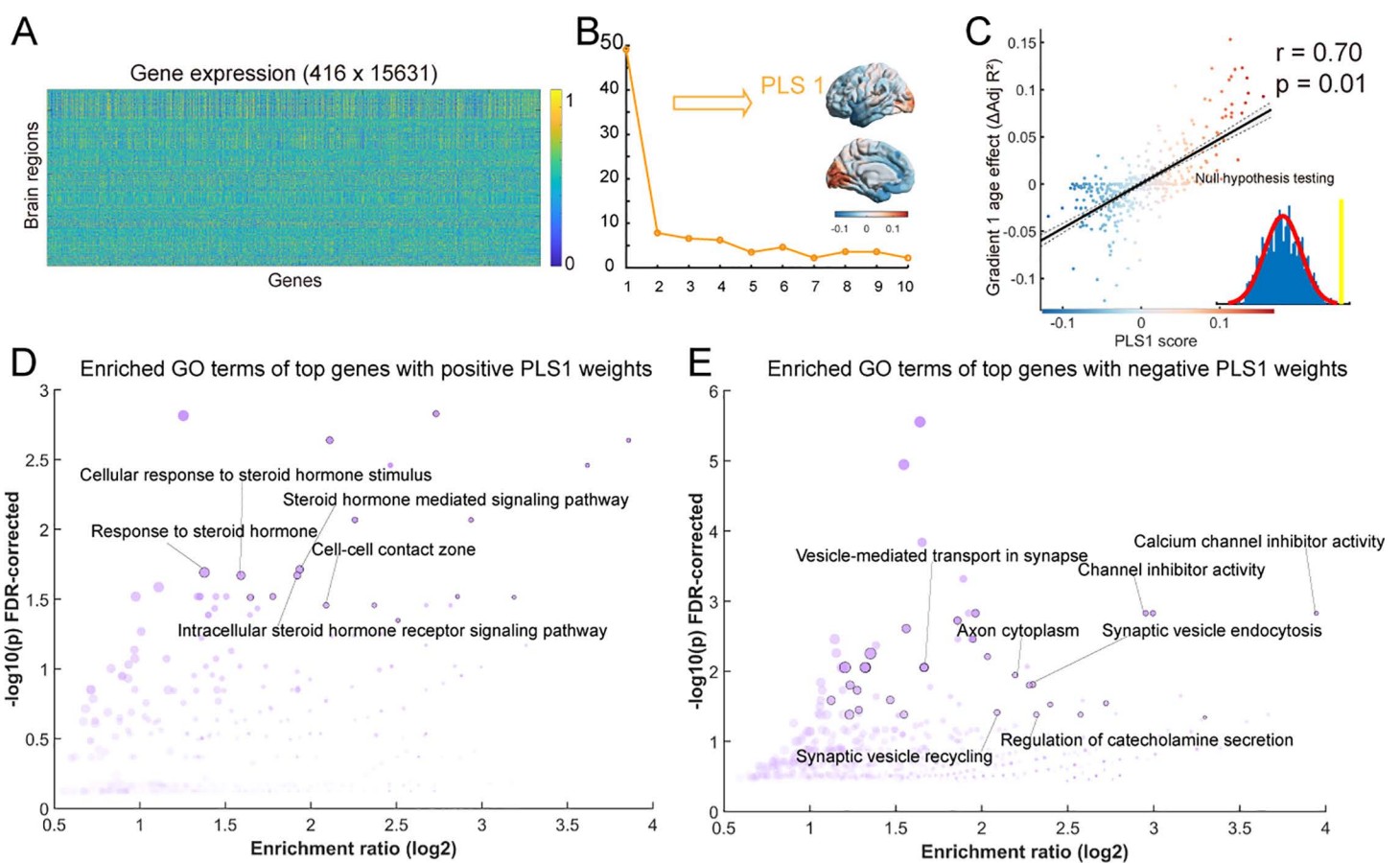

**Fig 6. Association between age-related changes in the principal gradient and gene expression profiles.** (A) Gene expression profiles across 416 brain regions. (B) The explained ratios for the first 10 components derived from the partial least squares regression analysis between age-related changes in the principal gradient and gene expression profiles. The first component (PLS1) accounted for the largest proportion of the variance and is depicted in the right panel. (C) Spatial correlation between age-related changes in the multiscale structural principal gradient and PLS1 scores. Each dot represents a brain region. The significance level was corrected for spatial autocorrelation ($p_{surrogate} = 0.01$). (D, E) Gene Ontology (GO) enrichment terms of the top 10% of genes with positive/negative PLS1 weights. The data underlying this figure can be found at https://zenodo.org/records/14874537.

(PLS1 −) weights were input into gene enrichment analysis using clusterProfiler package (all $p_{FDR} < 0.05$) [53]. Notably, the expression of positively weighted genes was positively correlated with the gradient 1 ΔAdj $R^2$-map. Gene Ontology (GO) analysis was performed to identify related molecular functions, biological processes, and cellular components. The significance of the GO enrichment terms was further evaluated using a null model, created by conducting enrichment analysis based on surrogate maps that preserved the empirically observed spatial autocorrelations [54]. As shown in Fig 6D, several meaningful puberty hormones-related terms emerged for the PLS1 + genes, such as "steroid hormone-mediated signaling pathway", "response to steroid hormone", "intracellular steroid hormone receptor signaling pathway" and "cellular response to steroid hormone stimulus" (Fig 6D). On the other hand, the PLS1-genes were enriched in several synapse-related terms, such as "vesicle-mediated transport in synapse", "synaptic vesicle endocytosis", "axon cytoplasm", and inhibition-related terms, such as "channel inhibitor activity", and "calcium channel inhibitor activity" (Fig 6E) (all $p < 0.05$, corrected for false discovery rate (FDR) and spatial autocorrelation). For a complete list of significant GO terms, refer to S12 Fig.

## Discussion

In this study, we documented the typical development process of multiscale structural gradients from childhood to adolescence based on an advanced structural connectome model. The results demonstrated that the maturation of a multiscale structural gradient was differentiated along the S-A cortical axis during the developmental period of 6–14 years of age. The shared developmental consequences of the multiscale structural gradient and cortical macrostructure indicated a potential interconnected maturation mechanism between the structural connectome and cortical morphology. The developmental changes in multiscale structure–function coupling reflected the refinement of functional specialization. In addition, the enhancement of the S-A axis pattern along the principal gradient demonstrated associations with enhanced cognitive performance and synapse-related gene expression. These findings provide a comprehensive understanding of the maturation principles of multiscale structural organization in the human brain during childhood and adolescence, as well as the underlying biological mechanisms involved.

### Differentiation of the multiscale structural principal gradient with development

The multiscale structural connectome model in this study integrated three complementary neuroimaging features, diffusion MRI tractography, MPC, and cortical GD [5]. Tract strength (TS) is the dominant measure for assessing white matter connectivity, while GD can infer short adjacent cortico-cortical connections [5,8]. MPC measures similarities between cortical regions, as connectivity is more likely to exist between regions with similar cytoarchitectures [10, 11]. Consistent with findings in healthy adults and adolescents aged 14–25 years [5,12], our study identified two principal axes of multiscale structural connectome organization, the primary-transmodal axis and anterior-posterior axis, in an accelerated longitudinal cohort aged 6–14 years. In this population, both qualitative (Fig 1D) and quantitative (Fig 2A, 2B, and 2C) analyses indicated an expanding gradient space during development that was mainly driven by the continuous differentiation of the principal gradient. Furthermore, given the more pronounced differentiation of the S-A axis, a primary-to-transmodal functional gradient was utilized as a proxy for this axis, and a tendency for the principal multiscale structural gradient to align with the S-A axis during development was revealed (Fig 2E). The continuous differentiation between the primary and transmodal cortex along the principal gradient aligned with the neurodevelopmental hierarchy from multiple findings, which suggested a varied developmental pattern between the primary and transmodal cortex [13]. First, this differentiation pattern along the principal structural gradient mirrored the increasing differentiation across the functional hierarchy during this period, as indicated by the shift in the principal functional gradient from the visual-sensorimotor gradient toward a pattern gradient characterized by the S-A axis [20, 21]. Second, this differentiation pattern was also consistent with evidence from white matter connectivity and myeloarchitecture, which demonstrated augmented differentiation of this axis during development [55, 56]. In addition, the differentiation of cortical features along the S-A axis may delineate distinct cognitive functions and facilitate executive, socioemotional, and mentalizing functions within the transmodal region [13]. Recent studies have indicated that differentiation along the S-A axis is related to flexible cognitive processing and better cognitive function [57, 58]. Our results corroborated this finding that a more differentiated gradient along the S-A axis was related to better cognitive performance, such as WM, attention performance, fluid intelligence episodic memory, and executive function.

## Interactions between the development of the multiscale structural gradient and cortical morphometric features

Our findings revealed coordinated spatiotemporal developmental patterns of cortical morphometric profiles that encompass multiple morphometric features and the principal multiscale structural gradient incorporating white matter and cortical microstructure. This observation corresponds with the notion that most aspects of brain development, including the associations between structural connectivity maturation and cortical morphology, follow a spatial pattern resembling the S-A axis [13]. In addition, from a more microscopic perspective, some empirical evidence and theoretical hypotheses have established associations between changes in cortical morphology and structural wiring; one hypothesis is Seldon's "balloon model" [59], which states that akin to an expanding balloon, the growth of white matter induces tangential stretching and thinning of its connected cortex. This hypothesis was supported by correlations found between cortical surface expansion and increased subcortical white matter fibers during development [34]. The theory proposed by Essen [35] links the cortical folding pattern to axonal mechanical tension, with gyri potentially formed through mechanical tension pulling closely interconnected regions together. Gray matter thinning during childhood and adolescence is attributed to biological processes, such as synaptic pruning, apoptosis [32,60], and proliferation of myelin at the interface between gray matter and white matter [31–33]. Previous studies also revealed associations between cortical thinning and increased white matter fibers during development [61, 62]. Furthermore, considering the brain's organization as a network of interconnected regions, a recent study adopting a network perspective demonstrated the constraints of the white matter network on the maturation of CT from childhood to adolescence [63]. Our study also revealed that regions exhibiting analogous structural connection profiles demonstrated congruent cortical morphology in spatial and maturation patterns, which can be elucidated through various mechanisms. First, structurally interconnected regions tend to possess similar cytoarchitecture and may develop during comparable time windows [64–66]. Regions with similar cytoarchitectonic patterns tend to exhibit similar morphological characteristics [29]. Second, the regionally heterogeneous developmental patterns of cortical morphology may be attributed to mutual trophic influences supported by structural wiring [67]. Third, a recent study demonstrated that regions with similar cytoarchitectonic features and white matter interconnections are more likely to exhibit similar neurotransmitter receptor profiles [68]. Consequently, these regions may be subject to coregulation through similar physiological mechanisms [63,69]. The findings of this study offer novel insights into the interconnected maturation mechanisms between cortical wiring and macrostructure, suggesting a potential role for structural connectivity in shaping cortical morphology.

## Relationships between changes in multiscale structural organization and functional organization during development

Our study revealed a continuous differentiation pattern along the principal multiscale structural gradient during development, paralleling the primary-to-transmodal functional gradient results reported by [21] in the same population as ours. This finding indicated a harmonized process of structural and functional maturation in human brain development, characterized by increasingly enhanced hierarchical organization and segregated topology. Previous studies also highlighted the synchronized maturation of structural and functional organization. A study based on functional intrinsic cortical activity revealed a hierarchical neurodevelopmental axis, which was linked to a progressive increase in intracortical myelination [70]. Moreover, throughout the developmental process, both the structural and functional topology

displayed a more distributed and segregated pattern [71, 72]. These results suggested a mature process of enhanced segregation, manifested in structural and functional synchronization.

In addition, numerous studies have consistently demonstrated that structure–function coupling exhibits regional heterogeneity, with the degree of coupling aligning along the S-A axis [50,73,74]. Our findings supported the prevailing trend, with a greater degree of coupling in the primary cortex than in the transmodal cortex. The primary regions exhibit more rapid and accurate responses to external stimuli, necessitating stronger structural constraints. In contrast, the transmodal regions are untethered from structural constraints, consistent with their more flexible and diverse functional roles [11,75]. Low coupling in the transmodal cortex may be related to functional flexibility and diverse task demands [76]. A previous study utilizing the white matter connectivity network and functional network demonstrated that their coupling reflects functional segregation [50]. Consistent with this study, our study also revealed a significant spatial correlation between multiscale structure–function coupling and the PaC, as well as their interrelated developmental patterns. Our findings revealed that during development, regions exhibiting stronger coupling between multiscale structural and FC demonstrated stronger functional specialization, characterized by a greater degree of segregation. Conversely, regions with weaker coupling showed a greater degree of integration. Notably, stronger coupling between structure and function supports faster and more accurate specialized functions, while regions with fewer structural constraints are associated with greater flexibility and integrative roles [11,75]. These results established a compelling connection between structural-functional coupling and the underlying mechanisms of cortical organization.

### Transcriptional profiling of the developmental multiscale structural gradient

Using gene expression data from the AHBA dataset, our transcriptome analysis revealed that developmental changes in multiscale structural gradient 1 were associated with the transcriptional profiles of genes involved in puberty hormones-, synapse-, and inhibition-related terms, such as "steroid hormone-mediated signaling pathway", "synaptic vesicle endocytosis", "axon cytoplasm", and "calcium channel inhibitor activity". During late childhood and adolescence, pubertal hormones like testosterone and estrogen significantly influence the development of cortical macrostructure and white matter structures [19]. Rodent studies show these hormones affect brain development, such as neuron pruning and cortical inhibition [77, 78]. Synapses serve as the foundation for communication between neurons in the nervous system. The elimination of synapses persists throughout development, with the pruning process exhibiting heterogeneity across brain regions and refining functional circuits [23,79]. Sensory regions complete this process during late childhood, while higher-order regions continue to experience synaptic pruning into adolescence [80]. Calcium ions trigger the release of neurotransmitters and initiate synaptic transmission [81]. Myelinated axons serve as the primary conduits for transmitting information within the central nervous system, constituting the majority of white matter. White matter pathways undergo continuous remodeling during brain maturation [82]. Moreover, combined with gene enrichment, previous studies on the development of functional networks, CT, and intracortical myelination have also reported associations with synapse-related terms [21,27,83,84]. Our findings may indicate possible synapse-related developmental process mechanisms underlying multiscale structural connectome development from childhood to adolescence.

### Limitations and future directions

There are several limitations to this study. First, our current dataset lacked pubertal hormone measurements, leading us to define ages chronologically instead of by pubertal stage. This

limitation may constrain our ability to investigate the effect of pubertal hormone levels on multiscale structural gradients. Incorporating pubertal-related measures into future analyses may yield significant biological insights. Second, dMRI tractography faces challenges in accurately reconstructing white matter connectivity, often resulting in false positives and negatives [85]. Specifically, the long-distance connections are often underrepresented in tractography results, as demonstrated by validation studies using tract-tracing methods [7]. Moreover, one of the greatest challenges lies in resolving complex fiber geometries, particularly in regions with intricate architectures where fibers cross, branch, or overlap [86]. Third, the gene-related analysis was a cross-dataset correlational inference, where gene expression profiles were derived from a small sample of adult postmortem brains (6 left hemispheres and 2 right hemispheres), potentially limiting its generalizability, especially to developmental stages. Furthermore, the use of microarray-based gene expression may have lower sensitivity and accuracy compared to RNA-Seq, which offers greater transcriptomic coverage and precision [87]. Future studies should incorporate pediatric-specific gene expression datasets with spatial resolution comparable to that of the AHBA, based on more precise transcriptomic measurement techniques. Notably, while we observe spatial correlations between typical cohort patterns and AHBA-derived gene expression, we emphasize that correlation does not imply causation. These limitations should be considered when interpreting our results.

## Materials and methods

### Participants

We performed analyses in two independent datasets. The discovery dataset was from the Children School Functions and Brain Development Project in China (Beijing Cohort), which contains a longitudinal dataset of 643 scans from 360 participants (163 females) aged 6–14 years. The final sample included 276 participants (aged 6–14 years, 135 females; 437 scans [159 for 1 time point, 83 for 2 time points, and 39 for 3 time points]) with complete, quality-controlled T1w and T2w images, dMRI scans, and rs-fMRI scans. All participants in this study were cognitively normal, and those with a history of neurological disorders, mental disorders, head injuries, physical illness, or contraindications for MRI were excluded. The replication dataset comprised a cross-sectional group of 290 participants aged 6–14 years, with 177 females and 113 males (sex self-reported), selected from the Lifespan HCP-D. In accordance with the Declaration of Helsinki, all or their parents/guardians gave written informed consent. For the CBD dataset, ethical approval was given by the Ethics Committee of Beijing Normal University (approval No. IRB_A_0004_2019001). For the HCP dataset, ethical approval was given by the Washington University Institutional Review Board (IRB#201204036).

### Data acquisition

**MRI acquisition.** For discovery dataset, high-resolution T1w MRI, diffusion MRI, and rs-fMRI data were obtained using 3T Siemens Prisma scanners at Peking University, Beijing, China. T2w scans were acquired using 3T Siemens Prisma scanners at HuiLongGuan Hospital, Beijing, China. The parameters of the T1w scans were as follows: repetition time (TR) = 2,530 ms; echo time (TE) = 2.98 ms; inversion time (TI) = 1,100 ms; flip angle = 7°; field of view (FOV) = 256 × 224 mm²; number of slices = 192; slice thickness = 1 mm; and bandwidth (BW) = 240 Hz/Px. The parameters of the T2w scans were as follows: 3D T2-SPACE sequence, TR = 3,200 ms, TE = 564 ms, acquisition matrix = 320 × 320, FOV = 224 × 224 mm², number of slices = 256, slice thickness = 0.7 mm, and BW = 744 Hz/Px. The rs-fMRI scans were acquired using an echo-planar imaging sequence with the following parameters: TR = 2,000 ms; TE = 30 ms; flip angle = 90°; FOV = 224 × 224 mm²; number of

slices = 33; number of volumes = 240; and voxel size = $3.5 \times 3.5 \times 3.5\,mm^3$. Diffusion MRI was performed using a high angular resolution diffusion imaging sequence with a 64-channel head coil with the following parameters: TR = 7,500 ms, TE = 64 ms, acquisition matrix = $112 \times 112$, FOV = $224 \times 224\,mm^2$, slices = 70, slice thickness = 2 mm, BW = 2,030 Hz/Px, and 64 diffusion-weighted directions ($b$-value = 1,000 s/mm$^2$) with 10 non-diffusion weighted b0 (0 s/mm$^2$). For replication dataset, the imaging acquisition details were available in [88].

**Behavioral data.** *Discovery dataset:*

WM test. WM is associated with complex tasks such as temporary storage and manipulation of information [89]. We used a numerical *N*-back task to estimate WM capacity [51]. Twelve blocks of tasks under three workload conditions—0-, 1-, and 2-back—were completed by participants. For the 0-back condition, participants were instructed to judge whether the current digit was 1. For the 1- and 2-back conditions, participants were asked to judge whether the current digit was identical to the previous one or two digits in the sequence. The d-prime index was computed for each condition to assess WM performance. The index was calculated as the inverse of the cumulative Gaussian distribution of the hit ratio subtracted by the inverse of the cumulative Gaussian distribution of the false alarm ratio. The detailed task design can be found in Hao and colleagues [51]. In this study, we included 365 data points.

Attentional test. Attention involves prioritizing task-relevant information processing while disregarding irrelevant information [89]. We used a child-friendly version of the Attention Network Test (ANT) [90] to evaluate attention performance, which was measured by the response time for the alerting, orienting, and EC tasks. The detailed task design can be found in Hao and colleagues [51]. We included 372 data points in our study.

*Replication dataset:*

We employed multiple cognitive measures including fluid intelligence, episodic memory, executive function/cognitive flexibility, inhibition, language/reading decoding, vocabulary comprehension, processing speed, and WM. Detailed task descriptions and scoring interpretations are available in the HCP-Aging Lifespan 2.0 Release documentation and on the NIH Toolbox website (https://www.nihtoolbox.org/domain/cognition/).

## MRI preprocessing

Structural and functional images from both datasets underwent the minimum preprocessing of HCP pipeline and were modified to fit the discovery dataset [91].

**Structural MRI.** For the discovery dataset, we performed anterior commissure-posterior commissure (AC-PC) alignment and brain extraction. Subsequently, the T1w and T2w images were coregistered using a rigid body transformation with a boundary-based registration cost function [92]. Then, the square root of the product of the T1w and T2w images was used to correct for the bias field [93]. These images were registered to the Chinese Pediatric Atlas (CHN-PD) [94]. Using FreeSurfer 6.0-HCP [95], cortical surfaces were generated in native space, and T2w images were used to refine the pial surfaces. Moreover, cortical ribbon volume myelin maps were generated [93].

**Diffusion MRI.** For the discovery dataset, diffusion images were initially preprocessed using MRtrix3 [96], which included denoising and removing Gibbs ringing artifacts [97]. Subsequently, the FSL eddy tool was employed to correct eddy current-induced distortions, head movements, and signal dropout [98–100]. Next, the eddy-corrected diffusion images and corresponding field maps were preprocessed using the FSL epi_reg script to effectively mitigate EPI susceptibility artifacts (https://fsl.fmrib.ox.ac.uk/fsl/fslwiki/FLIRT/UserGuide#epi_reg). The diffusion images were finally corrected for B1 field inhomogeneity using the N4 algorithm provided by ANTs [101]. Detailed information on the dMRI preprocessing steps can be found in [63]. For the replication dataset, we utilized the same

standard preprocessing steps in the MRtrix3 software as those used for the discovery dataset. The only exception is that the EPI distortion was corrected using the TOPUP tool in MRtrix3, as the HCP-D dataset included paired phase-encoded field maps.

**Functional MRI.** For discovery dataset, each frame of the functional time series was registered to the first frame using rigid body registration to correct for head motion. The distortions in the phase encoding direction were corrected using the corresponding field map. The first frame was subsequently registered to the T1w image using rigid body and boundary-based registrations to correct for distortions. The relevant transformations were concatenated to register each frame of functional time series to the first frame, native T1w space, and finally the CHN-PD atlas space. Then, bias field correction, extraction of the brain, and normalization of the whole-brain intensity were performed. Next, followed by a bandpass filter (0.01 Hz < $f$ < 0.08 Hz), we performed ICA-based Automatic Removal Of Motion Artifacts (ICA-AROMA) for denoising [102]. We also removed the shared variance between the global signal and time series. Subsequently, the time series in the CHN-PD volume space was projected onto native cortical surfaces using a partial volume weighted ribbon-constrained mapping algorithm. Next, the signals on the cortical surface were resampled and precisely aligned with the Conte69 template through registration, followed by resampling onto the fsaverage5 surface.

## Generation of multiscale structural features

Consistent with the previously reported multiscale model, three complementary structural features were calculated based on T1w, T2w, and diffusion images. The three features were mapped onto Schaefer 1000 parcellations and calculated as described below (the Schaefer 400 atlas was used for validation analysis) [46].

1. **Geodesic distance.** Based on the mid-thickness surface of the individual native surface, the GD was calculated as the shortest distance between two nodes along the surface. Consistent with the previous study, we first calculated the Euclidean distances from all vertices to each other and identified the vertex with the minimum total distance, designated as the centroid vertex for each parcel [5]. Next, we matched each centroid vertex to its nearest voxel in volumetric space. We then applied Chamfer propagation using the imGeodesics Toolbox (https://github.com/mattools/matImage/wiki/imGeodesics) to calculate distances to all other voxels while traversing through a gray/white matter mask [103]. This method enables the estimation of inter-hemispheric projections [5], unlike previous approaches that focused solely on intra-hemispheric distances [8,14]. Finally, we projected the distance estimations back from volumetric to surface space, averaged them within each node, and created a symmetric distance matrix.

2. **Microstructure profile covariance.** According to a previously reported protocol, we acquired 12 equivolumetric surfaces between the pial and white surfaces and sampled T1w/T2w values along the vertices of these surfaces [11]. The intensity profiles of T1w/T2w images were averaged within parcels, excluding any outlier vertices. Then, we calculated pairwise Pearson product-moment correlations between the intensity profiles of each pair of parcels while controlling for the average whole-cortex intensity profile. The matrix was log-transformed after thresholding at zero, resulting in the final MPC matrix.

3. **Tract strength.** We used MRtrix3 to generate a white matter connectivity network. For the discovery dataset, we registered T1w images and their corresponding data to the native diffusion MRI space. An unsupervised algorithm was used to estimate response function (RF) in different brain tissue types [104]. Then, we performed single-shell 3-tissue constrained

spherical deconvolution (SS3T-CSD) [105] using MRtrix3Tissue (https://3Tissue.github.io), a branch of MRtrix3 [96], to obtain the fiber orientation distribution in all voxels. Following intensity normalization, we chose the gray matter/white matter boundary as the streamline seed mask. Based on anatomically constrained tractography (ACT) [106] with the segmentation results of the structural MR images, second-order integration over fiber orientation distributions was employed to generate streamlines [107]. Streamline generation was terminated when 20 million streamlines were counted (maximum tract length = 250 mm; fractional anisotropy cutoff = 0.06; angle threshold = 45°). The spherical deconvolution-informed filtering of tracks approach was used to correct the bias of streamline density [108]. The TS was measured by the number of streamlines. Finally, white matter connectivity was generated by mapping the streamlines onto the Schaefer 1000 atlas and log-transformed. For the replication dataset, we followed the same procedures in the MRtrix3 software as with the discovery dataset. The only exception was that we employed multi-shell multi-tissue spherical deconvolution to derive the fiber orientation distribution [109]. Streamline generation was terminated when we reached a count of 5 million streamlines to optimize computational efficiency.

## Calculation of multiscale structural gradients

We used the BrainSpace Toolbox to compute connectome gradients (https://github.com/MICA-MNI/brainspace) [110]. Consistent with a previous study, the nonzero values of the MPC, TS, and inverted GD matrices were rank normalized and rescaled to the same numerical range [5]. The three matrices were horizontally concatenated and subjected to a diffusion map embedding algorithm with a kernel of normalized angle similarity, which mapped the high-dimensional multiscale structural connectome data into a low-dimensional space [111]. The distances in the gradient space reflect dissimilarities in connectivity patterns between regions. In line with previous studies, we set parameter $\alpha = 0.5$. By dividing the population into six groups based on age with 1-year intervals, we generated a group-level multiscale connectome by averaging the individual multiscale matrices. To make the gradients comparable across individuals and eliminate the randomness of the direction of the gradients, we used Procrustes rotations to align the individual gradients to their corresponding age-specific group-level gradients derived from the group-level multiscale connectome [112].

To examine the contribution of each structural connectivity feature to the multiscale structural gradients, we employed the following steps. We calculated gradients based on the matrix of each individual structural connectivity feature, including TS, GD, and MPC, and correlated them with the multiscale gradients, using partial correlation coefficients as a measure of contribution, while controlling for the influence of the other features. To focus our analysis, we selected the top three gradients at the group level for quantification. Recognizing the importance of accounting for spatial autocorrelation in brain maps, we generated 1,000 surrogate maps that preserved the spatial autocorrelation of the given brain map using the variogram matching approach to estimate the significance [54] (S1 Fig). S1 Fig shows the partial correlations between the first three gradients of TS, GD, and MPC with the multiscale structural gradients, indicating that MPC and GD significantly contribute to the multiscale structural principal gradient, while TS contributes relatively less.

To examine which structural connectivity features drive the development of the multiscale structural gradients, we performed same analysis based on the age-effect maps of the first three multiscale structural gradients and individual feature gradients. As shown in S7 Fig, the development of MPC and GD significantly contributes to the development of the multiscale structural principal gradient, while TS contributes relatively less to the multiscale structural principal gradient.

The global gradient measures were computed to summarize the age-related changes in the gradients. These global measures included the following: (1) gradient range, calculated as the difference between the maximum and minimum values; (2) explanation ratio, calculated as the eigenvalue divided by the sum of all eigenvalues; (3) standard deviation, defined as the standard deviation of the given gradient; and (4) gradient dispersion, calculated as the sum of the Euclidean distances of each node to the centroid in the 2D gradient space constructed by the first and third gradients. Moreover, we calculated the eccentricity measure as the Euclidean distance between each node and the centroid of the template space obtained from averaging the multiscale matrix across all participants.

## Correlation analysis with cortical morphometric features

To investigate the relationships between multiscale structural gradients and cortical morphometric features, we utilized cortical morphometric features derived from the results of the FreeSurfer preprocessing procedure. Subsequently, five cortical morphometric features, cortical thickness, gray matter volume, surface area, MC, and GC, were extracted and mapped onto the Schaefer 1000 atlas. Given the similarities of cortical patterns across these metrics, we performed PCA to generate a concise representation of the morphometric features. Specifically, for each participant, we conducted PCA on matrix X of node×feature. The first component captured the largest variance, and areas with similar morphological profiles were in close proximity along this principal axis. We conducted a correlation analysis between the first principal component and the multiscale structural gradient.

Additionally, consistent with previous studies, we constructed MSN for each subject based on five morphometric features [29,113]. Subsequently, we conducted a correlation analysis between MSN edges and multiscale structural connectivity edges. We then calculated MSN gradients using the diffusion map embedding technique and correlated the first MSN gradient with the multiscale structural principal gradient [111,113].

## Calculation of the functional gradient

To assess how structure supported the maturation of functional organization, we related multiscale structural gradients to the FC network. Considering the primary-transmodal functional gradient as a representative of the functional hierarchy and its gradual maturation throughout development, we conducted correlation analysis between structural gradients and functional gradient. We computed pairwise Pearson's correlation coefficients based on time series with the Schaefer 1000 atlas to obtain individual FC matrices, followed by the generation of a group-averaged FC matrix. We retained the top 10% of edges per row and computed the row-wise normalized angle similarity. This matrix was then input into the diffusion map embedding algorithm, yielding the primary-transmodal functional gradient [111].

## Analysis of multiscale structure–function coupling

We investigated multiscale structure–function coupling during youth, calculated as the Spearman rank correlation between structural connectivity and FC profiles at the nodal level. We computed the average of these individual maps across all participants to generate an averaged coupling map. To quantify the functional specialization of brain networks, we computed the PaC for each scan using the Brain Connectivity Toolbox (https://sites.google.com/site/bctnet/) [114,115]. Based on the Yeo functional networks [47], the PaC measured inter-module connectivity and quantified the extent to which a node participated in other modules.

In addition, consistent with the previous study [12], we constructed a wiring space based on the first and third multiscale gradients and calculated the Euclidean distance between

nodes as a measure of multiscale structural differentiation. Then, we computed the average differentiation and FC within and between Yeo functional systems.

## Statistical analysis

We employed GAMM with penalized splines in R (version 4.3.0) [116] utilizing the mgcv package [117] to explore the relationship between brain variables and age. This model incorporated age as a smooth term, subject ID as a random effect, and sex and mFD as linear covariates. We employed cubic splines for the smooth term, offering computational efficiency and appropriateness for data sparsity at boundaries. The restricted maximum likelihood approach was used for the optimization of smoothing parameters [118]. The GAMM smooth term for age produces a spline, modeled by weighted basis functions, that represents a developmental trajectory for each metric. To prevent overfitting of the spline, the maximum basis complexity ($k$) was set to 3, as the Bayesian information criterion (BIC) comparison indicated lower BIC values for most global gradient measures at $k = 3$ rather than $k = 4$. We assessed the effect size of each age spline by calculating the change in adjusted $R^2$ ($\Delta$Adj $R^2$) between the full GAMM and a reduced model excluding the age term. Statistical significance was evaluated using analysis of variance to compare the full and reduced models. Since $\Delta$Adj $R^2$ is an effect size and directionless, we extracted the sign of the age coefficient from the equivalent linear model and assigned it to $\Delta$Adj $R^2$.

This model was utilized to characterize developmental effects for global gradient measures, regional gradient scores, and other brain variables, and we further plotted the developmental trajectory of the principal gradient for each region. For the replication cross-sectional dataset used for validation, we removed the random effect term, keeping the rest unchanged.

For all spatial correlation analyses between different cortical maps, we used the variogram matching approach to estimate the significance [54]. By generating 1,000 surrogate maps that preserved the spatial autocorrelation of the given brain map, we repeated the correlation analysis utilizing these surrogate maps. The resulting correlation coefficients generated a null distribution comprising 1,000 values. The $p$-value was calculated as the proportion of the surrogate coefficients exceeding the actual coefficient.

## Gene enrichment analysis

We collected genome expression data from the AHBA to identify genes associated with age-related multiscale structural gradient changes (https://human.brain-map.org [52]). The AHBA is a regional microarray transcriptomic dataset of 3,702 tissue samples from six healthy adult donors. We used the abagen toolbox (version 0.1.3; https://github.com/rmarkello/abagen) to preprocess the microarray data using the Schaefer 1000 atlas. Using the default parameters, we finally obtained a 416×15,631 (region× gene) matrix. To be more specific, the microarray probes were reannotated according to data provided by [119], and probes not matched to a valid Entrez ID were discarded. Probes whose expression intensity exceeded background noise in 50% of samples across all donors were retained [120]. For genes with multiple probes, the probe with the most consistent pattern of regional variation across six donors was selected as representative probe, resulting in 15,631 genes. Then samples were assigned to brain regions based on their MNI coordinates, with strict constraints on assignment: samples had to be within 2 mm of a parcel and were constrained by hemisphere and structural division. We used Schaefer 1000 volumetric atlas in MNI space. Within-donor normalization across donors was performed to mitigate donor-specific effects. Finally, samples were averaged first within regions for each donor, then across donors to create the regional expression matrix. Given that right hemisphere data were only available from two donors, we opted to utilize the data from the left hemisphere for our analysis.

To determine the relationships between age-related changes in the multiscale structural gradient and genes, we used the previously obtained age-effect $\Delta$Adj $R^2$ statistics of the principal gradient ($\Delta$Adj $R^2$-map) and gene expression matrix in PLSR. Our goal was to identify the components associated with the gradient $\Delta$Adj $R^2$-map, which represented optimally weighted linear combinations of expression patterns. The first component (PLS1) was the most strongly correlated with the $\Delta$Adj $R^2$-map. By using a previously described spatial autocorrelation correction approach, we examined the statistical significance of the variance explained by the PLS components and the correlation coefficient between PLS1 and the $\Delta$Adj $R^2$-map [54]. Subsequently, bootstrapping was performed to assess the error of each gene weight from PLS1, and we transformed the weights into $Z$ scores by dividing the weight by the standard deviation of the given weight derived from 1,000 bootstrapping results. We selected the top 10% of genes from both the positive and negative weights, which made the largest contribution to PLS1, for the subsequent gene enrichment analysis.

The genes with positive and negative PLS1 weights were then separately entered into gene enrichment analysis using clusterProfiler package [53]. GO analysis was used to search for specific molecular function, biological process, and cellular component terms. Next, we adopted a null model approach, recognizing that spatial autocorrelation significantly drives false-positive bias in gene enrichment analysis. Specifically, we created 1,000 surrogate age-effected maps of the multiscale structural principal gradient using the variogram matching method to maintain spatial autocorrelation [54]. We then utilized PLS regression to identify the most relevant genes for each surrogate map, generating 1,000 null gene sets. Following this, we carried out GO enrichment analysis on each surrogate gene set, producing a null distribution of surrogate $p$-values for each GO term. Finally, we assessed the empirical GO term $p$-values against this null distribution, considering a result significant only if it fell below 95% of the surrogate $p$-values.

## Analysis of the relationship between cognition and the principal multiscale structural gradient

We performed PLSC analysis [121] with the myPLS toolbox (https://github.com/danizoeller/myPLS) to extract the relationships between the multiscale structural gradient and cognitive scores. PLSC analysis was performed separately for WM and attention performance. We first computed a covariance matrix $R$ between brain variables $X$ and cognition variables $Y$:

$$R = Y^t \times X \tag{1}$$

followed by singular value decomposition on $R$:

$$R = U \times S \times V^T \tag{2}$$

where $U$ and $V$ reflect the contributions of the cognition and brain variables, respectively, to the LCs, while $S$ represents the singular values. Then, brain scores ($Lx = X \times V$) and cognition scores ($Ly = Y \times U$) were computed for each LC by projecting brain and cognition variables onto their corresponding weights. Brain loadings and cognition loadings were computed as Pearson correlations between the original data and previously obtained scores. Overall, the PLSC analysis generated LCs that represented the optimal weighted linear combinations of the original variables, thereby establishing the strongest relationships between the brain and cognition data. Subsequently, we assessed the statistical significance of each LC using a permutation test ($n = 1,000$). Specifically, we randomly shuffled the cognitive data across all subjects, resulting in a null distribution of singular values. By comparing the actual value with the null distribution, we ascertained the statistical

significance. The statistical significance of brain and cognition loadings was estimated by bootstrap resampling ($n = 1,000$), with replacement across all subjects on $X$ and $Y$.

Due to the issue of optimistic within-sample effect size estimates in PLSC analysis, as highlighted by Marek and colleagues [122] and Helmer and colleagues [123], we implemented a cross-validation procedure. We conducted 10 iterations of 5-fold cross-validation, splitting our sample into training (80% of subjects) and testing (20% of subjects) sets for each iteration. The process for each fold involved performing PLSC analysis on the training set to obtain weights, which were then applied to the original data of the testing set for both brain ($X$) and behavioral ($Y$) measures to compute composite scores. We calculated the correlation ($r$-value) between the brain and behavior composite scores in the testing set. Statistical significance was determined by permuting the behavioral data 1,000 times for each fold.

## Supporting information

**S1 Fig. Correlations between individual feature gradients (MPC, TS, GD) and multiscale structural gradients.** The $x$-axis represents the top three gradients of MPC, TS, and GD, while the $y$-axis represents the partial correlations with multiscale structural gradients. The top, middle, and bottom plots correspond to the top three multiscale structural gradients, respectively. Red dots indicate actual correlation coefficients. Spatial autocorrelation was corrected using 1,000 surrogate maps generated via the variogram matching approach [54]. Box plots and density plots show the null distribution of 1,000 correlation coefficients. The data underlying this figure can be found at https://zenodo.org/records/14874537.
(TIF)

**S2 Fig. The first three multiscale structural gradients projected onto the cortical surface for each group.**
(TIF)

**S3 Fig. Developmental pattern of the third multiscale structural gradient. (A)** The third gradient on the cortical surface. **(B)** Global density map of the third gradient for each group. **(C)** Radar plot of the third gradient for comparison between 6–7 years group and other groups based on Yeo functional networks (left) [47] and laminar differentiation parcellation (right) [48]. **(D)** Correlation coefficient between the third structural gradient and primary-to-transmodal functional gradient changed across age. **(E)** The spatial pattern of age-effect on the third gradient, with the age effect quantified using $\Delta$Adj $R^2$. The results that survived Bonferroni correction are circled by black lines (Bonferroni corrected $p < 0.05$). The data underlying this figure can be found at https://zenodo.org/records/14874537.
(TIF)

**S4 Fig. Multiscale structural gradients during childhood and adolescence based on Schaefer 400 atlas. (A)** The group-level gradients based on the Schaefer 400 atlas. **(B)** Global density map of the principal gradient for each group. **(C)** Radar plot of the principal gradient for comparison between 6–7 years group and other groups based on Yeo functional networks [47]. **(D)** The development trajectory of gradient scores for each region. **(E)** The spatial pattern of age-effect on the principal gradient, with the age effect quantified using $\Delta$Adj $R^2$. The results that survived Bonferroni correction are circled by black lines (Bonferroni corrected $p < 0.05$). **(F)** The first and third structural gradients mapped into a 2D gradient space for 6–7, 9, and 12–13 years old group demonstrated an expansion pattern during development. **(G)** Global measures of the principal gradient changed across age groups, including the explanation ratio (left), range (middle), and standard deviation (right) (age-effect $p < 0.01$).
(TIF)

**S5 Fig. Independent replication of multiscale structural gradients during childhood and adolescence based on Schaefer 400 atlas.** The same steps were conducted to regenerate the multiscale structural gradients using an independent HCP-D dataset aged 6–14 years old. (TIF)

**S6 Fig. Developmental pattern of the second multiscale structural gradient. (A)** The second gradient on the cortical surface. **(B–D)** Global measures of the principal gradient changed across age groups, including the explanation ratio (B), range (C), and standard deviation (D) (age-effect are not significant). **(E)** Correlation coefficient between the second structural gradient and primary-to-transmodal functional gradient changed across age. **(F)** The spatial pattern of age-effect on the second gradient, with the age effect quantified using $\Delta$Adj $R^2$. (TIF)

**S7 Fig. Correlations between developmental effects of individual feature gradients (MPC, TS, GD) and multiscale structural gradients.** The *x*-axis represents the developmental effects of top three gradients of MPC, TS, and GD, while the *y*-axis represents the developmental effects of multiscale structural gradients. The top, middle, and bottom plots correspond to the top three multiscale structural gradients, respectively. Red dots indicate actual correlation coefficients. Spatial autocorrelation was corrected using 1,000 surrogate maps generated via the variogram matching approach. Box plots and density plots show the null distribution of 1,000 correlation coefficients. The data underlying this figure can be found at https://zenodo.org/records/14874537. (TIF)

**S8 Fig. Independent replication of association between the multiscale structural principal gradient and morphometric features.** Results of the analysis of relationships between cortical morphological features using the independent HCP-D dataset for ages 6–14, employing the same steps as in the discovery dataset. (TIF)

**S9 Fig. Association between the multiscale structural principal gradient and MSN gradient. (A)** The MSN was fed into diffusion map embedding algorithm. The first gradient captures the largest proportion of the variance. The group-averaged gradients were projected onto the cortical surface and visually represented (right). **(B)** The correlation between MSN edges and multiscale structural connectivity edges. **(C)** Spatial correlation between the multiscale structural principal gradient and MSN gradient 1. Each dot represents a brain node. The significance level was corrected for spatial autocorrelation ($p_{surrogate} = 0.00$). **(D)** Spatial correlation between age-related $\Delta$Adj $R^2$-maps of the multiscale structural principal gradient and MSN gradient 1 ($p_{surrogate} = 0.00$). **(E)** The spatial pattern of age-effect on the principal gradient, with the age effect quantified using $\Delta$Adj $R^2$. The results that survived Bonferroni correction are circled by black lines (Bonferroni corrected $p < 0.05$). (TIF)

**S10 Fig. Independent replication of multiscale structure–function coupling during development.** Results of the analysis of multiscale structure–function coupling using the independent HCP-D dataset for ages 6–14, employing the same steps as in the discovery dataset. (TIF)

**S11 Fig. Independent replication of PLSC analysis between the principal gradient and cognitive scores. (A)** Pearson correlations between the composite scores of the principal gradient and composite scores of cognitive measures. The inset figure shows the null distribution of singular values estimated by the permutation test ($n = 1,000$). **(B)** Loadings of cognitive

measures were calculated by Pearson correlation between the cognitive measurements and their composite scores. The shadows represent significant loadings tested by bootstrap resampling ($n = 1,000$). Cognitive measures included fluid intelligence, episodic memory, executive function/cognitive flexibility, inhibition, language/reading decoding, vocabulary comprehension, processing speed, and working memory. **(C)** Gradient loadings were calculated by Pearson correlation between gradient 1 and their composite scores. The loadings of regions with black lines were subjected to a significance test by bootstrap resampling ($n = 1,000$). The data underlying this figure can be found at https://zenodo.org/records/14874537.
(TIF)

**S12 Fig. Association between age-related changes in principal gradient and gene expression profiles. (A)** Complete list of significant GO enrichment terms of top 10% genes with positive PLS 1 weights. **(B)** Complete list of significant GO enrichment terms of top 10% genes with negative PLS 1 weights (all $p < 0.05$, corrected for FDR and spatial autocorrelation).
(TIF)

**S13 Fig. Association between functional connectivity and structural differentiation. (A, B)** The functional connectivity matrix/multiscale structural differentiation matrix was summarized according to Yeo functional networks [47]. **(C)** Correlation between multiscale structural differentiation and functional connectivity. **(D, E)** Age-effect on functional connectivity/multiscale structural differentiation, with the age-effect quantified using $\Delta$Adj $R^2$ and significant network reported with an asterisk (Bonferroni corrected $p < 0.05$). **(F)** The correlation between age effects on multiscale structural differentiation and age effects on functional connectivity.
(TIF)

**S1 Movie. The developmental process of the gradient space across different ages in discovery dataset.**
(GIF)

**S2 Movie. The developmental process of the gradient space across different ages in replication dataset (Schaefer 400 atlas).**
(GIF)

**S1 Table. The cross-validation results of partial least square correlation (PLSC) analysis.**
(DOCX)

## Acknowledgments

The authors would like to thank all the families and children for their support and participation.

## Author contributions

**Conceptualization:** Yirong He, Shuyu Li.

**Data curation:** Weiwei Men, Yanpei Wang, Daoying Wang, Mingming Hu, Zhiying Pan, Haibo Zhang, Ningyu Liu, Shuping Tan, Jia-Hong Gao, Shaozheng Qin, Sha Tao, Qi Dong, Yong He.

**Formal analysis:** Yirong He, Debin Zeng, Qiongling Li, Lei Chu, Xiaoxi Dong, Xinyuan Liang, Lianglong Sun, Xuhong Liao, Tengda Zhao, Xiaodan Chen, Tianyuan Lei.

**Funding acquisition:** Shuyu Li.

**Investigation:** Yirong He.

**Visualization:** Yirong He.

**Writing – original draft:** Yirong He.

**Writing – review & editing:** Debin Zeng, Qiongling Li, Lei Chu, Xiaoxi Dong, Yong He, Shuyu Li.

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
