## [Editor Report · Decision Letter 0]

12 Jun 2024

Dear Dr Li,

Thank you for submitting your manuscript entitled "The continuous differentiation of multiscale structural gradients from childhood to adolescence correlates with the maturation of cortical morphology and functional specialization" for consideration as a Research Article by PLOS Biology.

Your manuscript has now been evaluated by the PLOS Biology editorial staff as well as by an academic editor with relevant expertise and I am writing to let you know that we would like to send your submission out for external peer review.

Once your full submission is complete, your paper will undergo a series of checks in preparation for peer review. After your manuscript has passed the checks it will be sent out for review. To provide the metadata for your submission, please Login to Editorial Manager (https://www.editorialmanager.com/pbiology) within two working days, i.e. by Jun 14 2024 11:59PM.

Kind regards,

Luke

Lucas Smith, Ph.D.

Senior Editor

PLOS Biology

lsmith@plos.org

---

## [Decision Letter · Decision Letter 1]

1 Aug 2024

Dear Dr Li,

Thank you for your patience while your manuscript "The continuous differentiation of multiscale structural gradients from childhood to adolescence correlates with the maturation of cortical morphology and functional specialization" was peer-reviewed at PLOS Biology. It has now been evaluated by the PLOS Biology editors, an Academic Editor with relevant expertise, and by several independent reviewers.

In light of the reviews, which you will find at the end of this email, we would like to invite you to revise the work to thoroughly address the reviewers' reports.

As you will see below, the reviewers find your manuscript interesting, but also have some major concerns. R1 would like you to provide the rationale behind your model, whereas R3 would like to see the incorporation of additional neuroimaging features. Furthermore R2 and R3 would like you to verify your result with an independent dataset.

Having discussed the reviews with the Academic Editor, we think you should address all comments from the reviewers. Furthermore, we would also like you do discuss the limitations of dMRI tractography, including poor correlation with ground truth tract tracing, the abundance of false positives and the inability to measure the density of the inter-areal network.

Given the extent of revision needed, we cannot make a decision about publication until we have seen the revised manuscript and your response to the reviewers' comments. Your revised manuscript is likely to be sent for further evaluation by all or a subset of the reviewers.

**IMPORTANT - SUBMITTING YOUR REVISION**

*Re-submission Checklist*

*Published Peer Review*

*PLOS Data Policy*

*Blot and Gel Data Policy*

Sincerely,

Suzanne

Suzanne De Bruijn, PhD,

Associate Editor

PLOS Biology

sbruijn@plos.org

Reviewer #1:

This manuscript examines the development of structural connectivity gradients in a combined cross-sectional and longitudinal sample of children and adolescents using linear mixed effects models. Multiple analyses are performed. I can appreciate the utility of characterizing the maturation of structural connectivity along the SA axis.

However, the conceptual basis of the main analysis is problematic and the manuscript could benefit from more precise language.

The analysis rests on the combination of three different measures: structural connectivity (SC) measured with diffusion MRI, geodesic distances (GD) between pairs of regions, and microstructural profile covariance (MPC). The aggregation of these measures is proposed to constitute a model of cortical wiring. This is highly debatable. GD and MPC are not estimates of wiring. GD simply measures the distance between areas. Taking the inverse will emphasize short-range pairs, which are already over-emphasized in diffusion MRI. The use of MPC follows from evidence that areas with similar cytoarchitecture are more likely to be connected. This is a probabilistic statement and there is no guarantee that such areas will in fact be connected. SC, GD, and MPC are thus entirely different properties, with different ranges of values and different conceptual underpinnings. The rationale for their combination lacks a strong foundation, which makes it difficult to interpret highly abstract gradient embeddings of the concatenated matrices. The relative contributions of each to the results are also unclear. At a minimum, I suggest that the authors (a) provide a more compelling rationale for this model (other than citing prior work); and (b) try to parse the contributions of SC, GD, and MPC separately.

The associations between SC maturation and cortical morphology follow logically from the observation that most aspects of brain development show a spatial pattern that resembles the SA axis (Sydnor et al. Neuron, 2021).

The analysis presented in lines 323-349 showing that SC-FC coupling follows the SA-axis has already been demonstrated by several groups and the findings of the analysis of the participation coefficient follow from this fact.

The PLS analysis should be cross-validated as within-sample effect size estimates are optimistic (Marek et al. Nature, 2022; Helmer, et al. Communications Biol, 2024).

The gene enrichment analysis approach used here has been shown to be biased (Fulcher et al. Nature Comms, 2021). Appropriate null models should be used for inference.

The manuscript would benefit from greater precision and clarity of language. For example:

Line 32- "reorganization of large-scale anatomical wiring" could mean anything. Please be specific

Line 37 - the use of GD, MPC, etc means that the model is not one of cortical "wiring"

Line 45 - why is multiscale used so often. Only one scale is considered - that defined by the parcellation.

Line 71 - it is unclear how the brain can "exhibit" "principles"

Line 75 - It is unclear what it means for "multiscale organization" to be "embedded within complex biological mechanisms"

Line 88 - GD does not quantify wiring cost. It is sometimes used as an estimate, but carries many assumptions with this usage.

Line 89 - MPC does not evaluate the strength of structural connections. It measures MPC, which may or may not be related to the probability of connectivity between two areas.

Line 92 - Please specific how or why these features "enhance" connectivity.

Line 98 - unclear usage of "feature transition"

Lines 173-174 - unclear what "progressively distributed toward both ends during development" means

Line 177 and several other places in the manuscript refer to increases/decreases of the principal gradient. I believe the authors are referring to increases/decreases in the range of values. The analysis are very abstract and the authors should be clear about what they mean; e.g., an expansion of the poles of the gradient? Increases for regions/networks that are not at the poles are relative to the poles; e.g., an increase in DAN values on its own is not interpretable. Its relative distance from one pole of the other is what should be interpreted.

Line 208 - it is unclear precisely what an explanation ratio quantifies.

Lines 209 - 210 - extremes and consistency of what?

Line 218 - the dominance of the principal gradient would be revealed by the variance explained.

Reviewer #2: identified himself as Boris Bernhardt

The continuous differentiation of multiscale structural gradients from childhood to adolescence correlates with the maturation of cortical morphology and functional specialization

The authors built multimodal connectomes based on SC, geodesic distance mapping and microstructural similarity and then derived gradients of these. Age related changes were assessed in a longitudinal dataset aged from 6-14 years. They observed that the gradients spanned SA and AP axes, and they observed age related expansion of gradient space. Findings were shown to coincide with changes in cortical morphology and the authors observed associations to functional and cognitive measures.

This is overall an interesting study, taking advantage of longitudinal data, an advanced model of cortical wiring, and integrative methods that bridge different connectivity metrics, structural MRI, as well as functional and behavioral markers. The work demonstrates a noteworthy expansion of multiscale gradient pattern in the transition from childhood to adolescence, in particular along the sensory-association axis, and thus represents a logical extension of prior studies assessing multiscale gradient reconfigurations in later developmental stages.

I only have a few comments and suggestions:

1) In the reporting of the results, please provide measures of effect sizes and t-values together with the p-values, for both significant and non-significant effects. Please consistently indicuate which findings were corrected for multiple comparisons and which ones weren't.

2) The expansion of the multiscale pattern in childhood is very interesting. Coyuld the authors perform supporting analyses to determine which of the constituent features drives the overall findings the most? I assume geodesic distance relationships to change least while MPC and SC are likely changing more in that age range.

3) Figure 3E - I assume findings are corrected for spatial autocorrelation? Please comment and/or adjust as you did for eg Figure 3C. Ditto for findings in F4 D and F4F. Would also co-plot the null distributions for these plots just to make presentation consistent.

4) What is the rationale to specifically assess WM and attentional ability. I understand these are domains that change but are the findings specific to those domains? A broader assessment of cognitive measures (if available in the dataset) could be worthwhile here to show which domains relate and which dont relate.

5) I wonder whether the gene expression correlations adds much to the paper, given that AHBA is mainly based on adults and that the sample size is small. These findings are of course not redundant, but the authors could consider moving those somewhat exploratory analyses to the supplements.

6) For independent verification, the anonymized raw and processed should ideally be made openly accessible, as longitudinal data in that age range is currently lacking. Alternatively, or in addition, the findings could be strengthened if results were replicated eg using the open access ABCD dataset (which has longitudinal data starting at 9 yo).

Reviewer #3:

This paper used multiscale structural architecture gradients to investigate the developmental process from childhood to adolescence (aged 6 to 14 years). The authors used an in vivo model of cortical wiring that combines features of white matter tractography, geodesic distance, and microstructural similarity to construct a multiscale brain structural connectome. Furthermore, the authors studied the relationships between the development process of multi-scale structural gradient space and cortical morphometric features, functional interactions, cognitive behavior, and gene expression profiles.

Overall, I would fully endorse the publication if the following major concerns can be addressed:

1. A recent study by Casey et al. (2020) proposed an in vivo structural wiring model to characterize comprehensive neural organizations across multiple scales. This model integrates various neuroimaging features to provide a detailed understanding of brain structure connectivity. Specifically, the authors focused on three key features: tract strength (TS) at the macroscale, cortical geodesic distance (GD), and microstructural profile covariance (MPC). Indeed, regarding the integration of additional features such as Morphometric Similarity Network (MSN) or Morphometric INverse Divergence (MIND), these methods introduced by Seidlitz et al. (2018) and Sebenius et al. (2023), respectively. Why did the authors use only three neuroimaging features to build an in vivo multi-scale structural wiring? What are the results of the development process of multi-scale structural gradients combining four neuroimaging features (i.e., TS, GD, MPC, MSN (or MIND))? Including MSN or MIND in analyses would likely enhance the model's ability, thereby enriching the brain's characterization of multiscale structural wiring.

2. The authors utilized workbench commands (-surface-geodesic-distance) to compute the distance of each pair of centroid vertices within the given parcel. However, given the limitation of this approach in calculating the geodesic distance (GD), only intrahemispheric distance. Calculating interhemispheric GD by averaging GD between the two hemispheres would introduce errors. A recently published GD calculation approach that combines surface and volume-based GD calculations. This approach rst matched each vertex to the nearest voxel in volume space. Then, the distance to all other voxels traveling through a gray/white matter mask was calculated using a Chamfer propagation. I strongly recommend that authors use this method to calculate geometric distance matrixes.

3. The authors performed PCA to generate a concise representation of the morphometric features (5 cortical morphometric features, CT, GMV, SA, MC, and GC). I found it bizarre that the physical significance of morphometric PC1.

Previous studies demonstrated that the principal gradient of MSN reveals diverged hierarchical organization in sensory-motor cortices (Yang Siqi, Cell Reports, 2021; Li Jiao, et al. Plos Biology, 2024). I am interested in the relationships between MSN gradient changes and multi-scale structure gradient changes from childhood to adolescence.

4. The authors investigated multiscale structure-function coupling during youth, which was calculated as the Spearman rank correlation between structural connectivity and FC profiles at the nodal level. Why do the authors not investigate the relationship between functional connectivity and multiscale cortical structural differentiation (i.e., the Euclidean distance in the wiring space)? It would be interesting to see the longitudinal structure-function coupling between age effects on functional connectivity and structural differentiation.

5. To quantify the effect of age on multiscale structural gradients during development, the authors constructed linear mixed-effect (LME) models. However, the development process is not always linear (Bethlehem Richard AI, et al. Nature 2022). A generalized additive model for location, scale, and shape (GALSS), a robust method for modeling nonlinear growth trajectories, would be useful for constructing the development process from childhood to adolescence.

6. The physical meaning of t-statistic of age-related changes is difficult to comprehend. To provide insight into the overall magnitude and direction of regional age effects, many previous studies (such as Sydnor Valerie, et al. Nature neuroscience, 2023; Li Jiao, et al. Plos Biology, 2024) have represented the age effects (R2 values) as the age-related regional changes during development. Therefore, I suggest replacing the t-statistic with the age effect (R2 values).

7. It would be necessary to use an independent data set (such as., HCPD, aged 6 to 14 years) to verify the repeatability of the results.

8. Given that right hemisphere data were only available from 2 donors, the authors opted to utilize the data from the left hemisphere for our analysis. Using the default parameters, a 416 ×15631 matrix was obtained. Why did the authors obtain a 416 ×15631 gene expression matrix instead of a 500 ×15631 matrix.

9. Abstract is somewhat redundant. Authors should simplify and improve abstract.

10. The full title of the manuscript is too long, and a short title is strongly recommended.

11. The Results section contains too much description of the Methods. The authors should mainly describe the results and what they reveal.

---

## [Decision Letter · Decision Letter 2]

6 Dec 2024

Dear Dr Li,

Thank you for your patience while we considered your revised manuscript "Multiscale structural gradient differentiation correlates with cortical morphology maturation and functional specialization from childhood to adolescence" for consideration as a Research Article at PLOS Biology. Your revised study has now been evaluated by the PLOS Biology editors, the Academic Editor and the original reviewers.

In light of the reviews, which you will find at the end of this email, we are pleased to offer you the opportunity to address the remaining points from the reviewers in a revision that we anticipate should not take you very long. We will then assess your revised manuscript and your response to the reviewers' comments with our Academic Editor aiming to avoid further rounds of peer-review, although might need to consult with the reviewers, depending on the nature of the revisions.

**IMPORTANT - SUBMITTING YOUR REVISION**

*Resubmission Checklist*

*Published Peer Review*

*PLOS Data Policy*

*Blot and Gel Data Policy*

Sincerely,

Christian

Christian Schnell, PhD

Senior Editor

PLOS Biology

cschnell@plos.org

REVIEWS:

Reviewer #1: The authors have address most of my comments. However, the new Fig S1 indicates that their model is mainly determined by MPC.

Since the MPC, TS, and GD maps are all correlated to some extent, the authors should examine partial correlations.

If it turns out that their model is largely driven by MPC, then the authors should demonstrate the benefits of their chosen model over just using MPC measures.

Reviewer #2: I thank the authors for their revisions. I only have few outstanding comments.

1) Please be more explicit in discussing limitations of using AHBA in the current work. Specifically, provide additional context that this analysis is a cross-dataset correlational inference, where patterns seen in typically developing cohorts are spatially associated to gene expression patterns seen in 6 left hemispheres and 2 right hemispheres of primarily adult donors. Please discuss some potential imitations of microarray gene expression analysis relative to eg RNASeq based approaches. Please emphasize the well known issue that correlation does not implicate causality necessarily.

2) As for my Q6, I would think PLOS guidelines mandate the open access of data that are used for the main analysis, so the authors would at minimum need to upload the used feature data to eg off so that other groups can replicate results.

Reviewer #3: The authors have addressed all issues. No further question now.

---

## [Editor Report · Decision Letter 3]

24 Jan 2025

Dear Dr Li,

Thank you for your patience while we considered your revised manuscript "Multiscale structural gradient differentiation correlates with cortical morphology maturation and functional specialization from childhood to adolescence" for publication as a Research Article at PLOS Biology. This revised version of your manuscript has been evaluated by the PLOS Biology editors and the Academic Editor.

Based on our Academic Editor's assessment of your revision, we are likely to accept this manuscript for publication, provided you satisfactorily address the remaining points raised by the Academic Editor (see below). Please also make sure to address the following data and other policy-related requests:

* We would like to suggest a different title to improve its accessibility for our broad audience: "The multiscale brain structural re-organization that occurs from childhood to adolescence correlates with cortical morphology maturation and functional specialization"

* Please include information whether your study has been conducted according to the principles expressed in the Declaration of Helsinki.

* Please provide the approval number(s) from the ethics committee(s).

* DATA POLICY:

Regardless of the method selected, please ensure that you provide the individual numerical values that underlie the summary data displayed in the following figure panels as they are essential for readers to assess your analysis and to reproduce it: 5BE, S1ABC, S7ABC and S11B.

* Please ensure that you are using best practice for statistical reporting and data presentation. These are our guidelines https://journals.plos.org/plosbiology/s/best-practices-in-research-reporting#loc-statistical-reporting and a useful resource on data presentation https://journals.plos.org/plosbiology/article?id=10.1371/journal.pbio.1002128

* CODE POLICY

We expect to receive your revised manuscript within two weeks.

*Published Peer Review History*

*Press*

Sincerely,

Christian

Christian Schnell, PhD

Senior Editor

cschnell@plos.org

PLOS Biology

Academic Editor remarks:

The paper needs to discuss the limitations of diffusion MRI tractography more carefully. The citations (refs 86 and 87) are out of date and do not include tract tracing validation studies. In the section entitled "Limitations and future directions" the difficulties facing dMRI tractograpy are understated. The Donahue et al., J.Neuroscience June 22, 2016 study shows that tractography fails for long-distance connections. Contrary to what the authors report, tractography does give a reasonable measure of stream line counts with respect to connection weights obtained from tract tracing.

Furthermore, in the Introduction the coverage of tractography is confusing and citation 7 is out of date.

---

## [Editor Report · Decision Letter 4]

19 Feb 2025

Dear Dr Li,

Thank you for the submission of your revised Research Article "The multiscale brain structural re-organization that occurs from childhood to adolescence correlates with cortical morphology maturation and functional specialization" for publication in PLOS Biology. On behalf of my colleagues and the Academic Editor, Henry Kennedy, I am pleased to say that we can in principle accept your manuscript for publication, provided you address any remaining formatting and reporting issues. These will be detailed in an email you should receive within 2-3 business days from our colleagues in the journal operations team; no action is required from you until then. Please note that we will not be able to formally accept your manuscript and schedule it for publication until you have completed any requested changes.

When you attend to those requests, please also double check that the zenodo links and Data and Code Availability statements are now correct.

PRESS

Sincerely, 

Christian

Christian Schnell, PhD

Senior Editor

PLOS Biology

cschnell@plos.org